# T cells translate individual, quantal activation into collective, analog cytokine responses via time-integrated feedbacks

**Karen E Tkach[1,2], Debashis Barik[1†], Guillaume Voisinne[1], Nicole Malandro[2,3], Matthew M Hathorn[1,2], Jesse W Cotari[1,2], Robert Vogel[1,2], Taha Merghoub[3], Jedd Wolchok[3], Oleg Krichevsky[4], Grégoire Altan-Bonnet[1,2]***

[1]Program in Computational Biology and Immunology, Memorial Sloan Kettering Cancer Center, New York, United States; [2]Graduate program, Weill Cornell Medical College, New York, United States; [3]Immunology Program, Memorial Sloan Kettering Cancer Center, New York, United States; [4]Physics Department, Ben Gurion University, Beer-Sheva, Israel

**Abstract** Variability within isogenic T cell populations yields heterogeneous 'local' signaling responses to shared antigenic stimuli, but responding clones may communicate 'global' antigen load through paracrine messengers, such as cytokines. Such coordination of individual cell responses within multicellular populations is critical for accurate collective reactions to shared environmental cues. However, cytokine production may saturate as a function of antigen input, or be dominated by the precursor frequency of antigen-specific T cells. Surprisingly, we found that T cells scale their collective output of IL-2 to total antigen input over a large dynamic range, independently of population size. Through experimental quantitation and computational modeling, we demonstrate that this scaling is enforced by an inhibitory cross-talk between antigen and IL-2 signaling, and a nonlinear acceleration of IL-2 secretion per cell. Our study reveals how time-integration of these regulatory loops within individual cell signaling generates scaled collective responses and can be leveraged for immune monitoring.

*For correspondence:
altanbog@mskcc.org

Present address: †School of Chemistry, University of Hyderabad, Hyderabad, India

Competing interests: The authors declare that no competing interests exist.

## Introduction

Understanding how collective biological function emerges from individual cell signaling remains challenging: rapid, binary decisions at the individual cell level (e.g., signal transduction and gene activation) must be bridged to graded, longer-term outcomes at the population level (e.g., proliferation, survival, and differentiation). This issue is particularly acute for the adaptive immune systems of metazoans, in which multicellular lymphocyte responses scale to the strength of pathogenic challenge across many spatial and temporal ranges. Investigations of information processing in mammalian systems have shown that regulations such as feedback (*Becker et al., 2010*) or population averaging (*Cheong et al., 2011*) are required to generate a large dynamic range of biological response from the limited resolution of proximal cell signaling.

Previous studies have demonstrated that T cell population responses to antigens vary widely with the quality and quantity of pathogenic stimuli several days after immunization (*Zehn et al., 2009*; *van Heijst et al., 2009*). Indeed, T cell receptors (TCRs) are highly sensitive to differences in antigenic signal strength within the first minutes and hours following contact with peptide-MHC (pMHC) complexes (*Lanzavecchia and Sallusto, 2001*). However, phenotypic variability within clonal populations of T cells results in heterogeneous sensitivity to shared antigen stimuli (*Feinerman et al., 2008*). Therefore, isogenic T cells in the same antigenic environment can display completely different signaling responses. Furthermore, early antigen discrimination is all-or-none (*Das et al., 2009*), with limited

**eLife digest** The cells of the immune system face the challenge of removing viruses and other pathogens without endangering healthy tissues. Cells called T cells plays a variety of roles in the immune response: some T cells directly destroy infected cells, some recruit other cells called phagocytes to the site of infection, and some release small proteins called cytokines. These cytokines help cells to communicate with other cells and, therefore, to tailor the overall immune responses to deal with a particular pathogen.

It is known that mammals are capable of adjusting the T cell response to match the overall severity of an infection. However, it is not clear how individual T cells coordinate their seemingly binary response—they are either activated when they recognize a pathogen, or they are not activated—into a response at the collective cell level that can be varied continuously over a wide range of values.

Here, Tkach et al. show that T cell populations match their production of the cytokine interleukin 2 (IL-2) to the abundance of antigens—molecules released by the pathogen—over an unexpectedly large range of concentrations. Through a combination of experimental and computational analyses, Tkach et al. identified two novel IL-2 feedback loops that help to generate the correct quantity of cytokine, irrespective of the total number of T cells. Furthermore, this model can be used to estimate antigen quantities within diseased tissues. The work of Tkach et al. illustrates the potential of feedback integration in cell signalling and gene regulation as a mechanism to allow cellular populations to respond to environmental stimuli in a graded, collective fashion.

dynamic ranges in both antigen input sensing and response output (*Altan-Bonnet and Germain, 2005*; *Tkach and Altan-Bonnet, 2012*; *Huang et al., 2013*), but long-term signaling can widen the gradation of functional responses. For example, a 100-fold difference in bacterial infection load is poorly resolved by a 1.5-fold change in the number of activated clones, yet ultimately results in a 20-fold shift in the magnitude of T cell expansion, notably as a result of sustained antigen signaling (*van Heijst et al., 2009*). How the noisy, bimodal decisions of single cells are coordinated to reflect global cognate antigen load over several days of response is currently unclear.

Paracrine cytokines that are secreted upon activation present an elegant solution for scaling population-level lymphocyte responses by translating individual antigen stimulation into a 'public good' on a longer time scale. Yet collective cytokine accumulation may be highly dependent on population size, as anticipated by quorum sensing systems (*Nealson, 1977*; *Dai et al., 2012*). Surprisingly, the antigen-scaling of in vivo clonal effector responses is mostly unaffected by large variability in the initial number of responding T cells (*Smith et al., 2000*; *Jenkins and Moon, 2012*). Thus, lymphocyte populations require mechanisms to create a wide dynamic antigen response range (*Becker et al., 2010*; *Tay et al., 2010*; *Cheong et al., 2011*; *Waysbort et al., 2013*), independently of initial population size (*Smith et al., 2000*; *Quiel et al., 2011*). It has been qualitatively proposed that competition for antigen may normalize for clonal density by limiting the duration of antigen signaling—and consequently, outcomes such as proliferation—within larger populations (*Smith et al., 2000*; *Tkach and Altan-Bonnet, 2012*). However, the quantitative molecular mechanisms required to compensate for hundreds-fold differences in population size within physiological timescales remain unknown.

In bridging the molecular, cellular, and population-level scales that regulate immune function, there is much to be gained from quantitative and theoretical approaches (*O'Garra et al., 2011*). Biological studies frequently apply genetic tools to dissect the nodes of regulatory networks. However, the ability to quantitatively track the route from molecular perturbation to functional phenotype requires an integrative and dynamic framework. Experimentally validated computational models have the capacity to generate quantitative predictions and establish the minimal requirements for the emergence of biological phenotypes. Moreover, experimental characterization and predictive modeling of the quantitative, dynamic relationships between system components can reveal regulatory architecture without genetic perturbation (*Yosef et al., 2013*). Such quantitative approaches have been successfully applied in varied studies of the immune system, from lymphocyte signaling (*Chakraborty et al., 2009*; *Das et al., 2009*) to receptor repertoire generation and thymic development (*Weinstein et al., 2009*; *Kosmrlj et al., 2010*; *Mora et al., 2010*; *Georgiou et al., 2014*), competition for cytokines

(*Busse et al., 2010*; *Feinerman et al., 2010*), lymphocyte differentiation (*Schulz et al., 2009*; *Francois et al., 2013*) and host-pathogen interactions (*Ciupe et al., 2007*; *Althaus and De Boer, 2008*).

Ultimately, quantitative analysis of immune responses can facilitate discovery in settings where gene modification is impractical, as in primary human cells, or where the direct and indirect effects of genetic alterations mask subtle or unanticipated interactions (*Sontag et al., 2004*). Although models always fail to capture the full complexity of immune responses, we posit that model building allows thorough, iterative interrogation of the sufficiency of molecular steps to account for large-scale functional properties (*Kemp et al., 2007*; *Janes et al., 2008*). Therefore, computational models are ideal for directly testing the emergence of collective responses from signaling within individual lymphocytes.

One candidate mediator of lymphocyte cooperation is interleukin-2 (IL-2), a paracrine cytokine produced and shared by activated T cells (*Smith, 1988*). Since IL-2 is secreted early after antigen challenge yet quantitatively tunes late decisions such as the magnitude of expansion and differentiation program of T cells (*Williams et al., 2006*; *Bachmann et al., 2007*; *Pipkin et al., 2010*; *Liao et al., 2011*; *McNally et al., 2011*; *Boyman and Sprent, 2012*), its accumulation may also link disparate time scales of cellular activation. Studies of IL-2 at single, early timepoints have reported that IL-2 scaling is limited in dynamic range, reflecting only the number of digitally activated T cells (*Podtschaske et al., 2007*; *Huang et al., 2013*). However, IL-2 production and consumption are modulated by several known feedbacks downstream of IL-2 signaling (*Smith, 1988*; *Long and Adler, 2006*; *Villarino et al., 2007*; *Boyman and Sprent, 2012*; *Yamane and Paul, 2012*), which could alter the dynamic range (*Nevozhay et al., 2009*; *Becker et al., 2010*) and T cell number-dependency of IL-2 output over time.

Here, we quantitatively characterize the cue-signal-response module of antigen-driven IL-2 secretion, and find that the empirical scaling of IL-2 accumulation challenges current understanding of this cytokine's production. Simulations of the known regulatory elements of the IL-2 pathway (*Feinerman et al., 2010*) predict a low saturating threshold and a strong population size dependence for IL-2 output. However, we demonstrate that IL-2 accumulation by T cells scales as a power law with antigen quantity, independently of population size, providing a shared quantitative readout of the global antigen load. Through experimental and computational probing, we uncovered two critical regulatory elements—a cross-talk interaction and a non-linear feedback—whose inclusion in the model captured the dynamics and scaling of collective IL-2 accumulation, and allowed for accurate prediction of the IL-2 pathway in a polyclonal system. Our study demonstrates how integration of feedbacks over long timescales enables variably sized populations of cells to respond proportionally to a large range of stimuli. Furthermore, these feedbacks carry information about the initiating TCR signal. Indeed, the observed cross-talk between TCR and IL-2 receptor signaling can be used to estimate the degree of antigen signaling experienced by activated T cells in response to un-calibrated stimuli such as explanted tumor tissue.

## Results

### Population size-independent scaling of IL-2 accumulation with antigen dose

We measured various input/output relationships for activation of different numbers of primary 5C.C7 TCR transgenic T cells responding to antigen presenting cells pulsed with varied doses of K5 antigen in vitro. In single timepoint snapshots, our observations aligned with previous work (*Altan-Bonnet and Germain, 2005*; *Podtschaske et al., 2007*; *Huang et al., 2013*), showing that T cells respond in a quantal manner to graded doses of antigen across various readouts (ERK phosphorylation, IL-2Rα or IL-2 expression, and cell cycle entrance—*Figure 1A*). Increasing the stimulating dose of antigen for 100,000 T cells did amplify the frequency of activation, but not in proportion to the shift in input stimulus: several readouts were saturated for higher antigen doses, and at best, a 1000-fold increase in antigen translated into 50-fold gain in the IL-2 production response (*Figure 1B*). IL-2 accumulation at 12 hr reflected this limited antigen-scaling in the frequency of IL-2 producers; furthermore, scaling at 12 hr was greatly influenced by the numbers of T cells in the system, with smaller populations demonstrating poorer antigen resolution (*Figure 1C*). However, probing further the IL-2 dynamics over days post-activation, we found that IL-2 accumulated rapidly and non-linearly with time, then exponentially decreased (*Figure 2A–B,D*), despite limited variation in the number of cells (*Figure 2C*). For each condition, we characterized these dynamics by their apex, $[IL-2]_{max}$, a quantity which was proportional to the T cell population's total [IL-2] accumulation over time (*Figure 2E*). Strikingly, $[IL-2]_{max}$ scaled as a

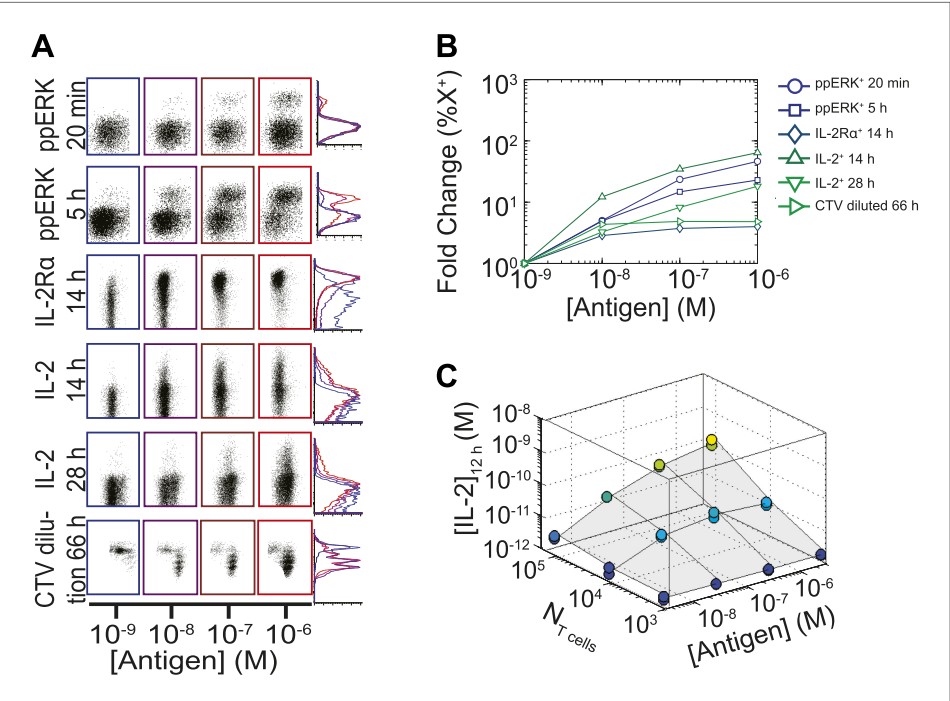

**Figure 1**. Limited dynamic range of T lymphocyte activation snapshots at the individual cell level. Varied numbers ($N_{Tcell}$) of 5C.C7 TCR Transgenic $Rag2^{-/-}$ T cells were cultured in duplicate in 200 µl of complete medium with $5.10^5$ I-E$^k$-expressing B10.A $Cd3e^{-/-}$splenocytes pulsed with varied concentrations of K5 peptide ([Antigen]). (**A** and **B**) Cells were harvested at varied timepoints and analyzed by flow cytometry for phosphorylation of ERK, upregulation of IL-2Rα, production of IL-2 via a Miltenyi IL-2 secretion assay, or dilution of Cell Trace Violet (CTV) upon cell proliferation. These measurements demonstrate (**A**) the bimodality of T cell activation as well as (**B**) the limited dynamics range of response for varied doses of antigens. (**C**) Supernatants were also collected at 12 hr and [IL-2] was measured by ELISA.

power law that could distinguish more than three orders of magnitude of input antigen dose (*Figure 2F*), despite the aforementioned saturated dynamic range of early T cell responses. Furthermore, [IL-2]$_{max}$ was essentially independent of the number of antigen-specific T cells in the culture ($N_{T cells}$) (*Figure 2G*). Given the same stimulus, 1000 T cells accumulated equal or greater amounts of IL-2 as 100-fold larger populations, in less than twice as much time. This could not be accounted for by differential changes in cell number (through proliferation or death), which were at most two-fold by the time parity was gained in IL-2 accumulation (*Figure 2C*). From 118 conditions over six independent experiments, we derived a simple empirical scaling law that summarizes the population size-independence and large scalability of IL-2 accumulation (*Figure 2H*):

$$[IL-2]_{max}^{experiment} \propto (N_{T cells})^{-0.10(\pm0.02)} \times [Antigen]^{+0.76(\pm0.05)}. \qquad (1)$$

This result challenged prior understanding of the extensively studied IL-2 pathway (*Figure 3A*). It has been established that T cells bimodally secrete IL-2 (*Podtschaske et al., 2007*) and express the α chain of the IL-2 receptor (IL-2Rα) (*Sheldon et al., 1993*) following TCR engagement. IL-2 production is subsequently switched off after loss of TCR signaling (*Huppa et al., 2003*) and/or gain of IL-2 response via phosphorylation of the transcription factor STAT5 into pSTAT5 (*Long and Adler, 2006*; *Villarino et al., 2007*; *Feinerman et al., 2010*; *Waysbort et al., 2013*), which also mediates further upregulation of IL-2Rα (*Smith and Cantrell, 1985*; *Waysbort et al., 2013*). Implementation of this classical model (*Figure 3B–E*) predicted that T cells would accumulate IL-2 commensurably with the size of their population. Furthermore, thiss model of IL-2 production predicted a weak antigen dependency that saturated near the canonical IL-2 signaling threshold of 10 pM (*Smith, 1988*) (*Figure 3E*), revealing a discrepancy between the established pathway regulation and our experimental

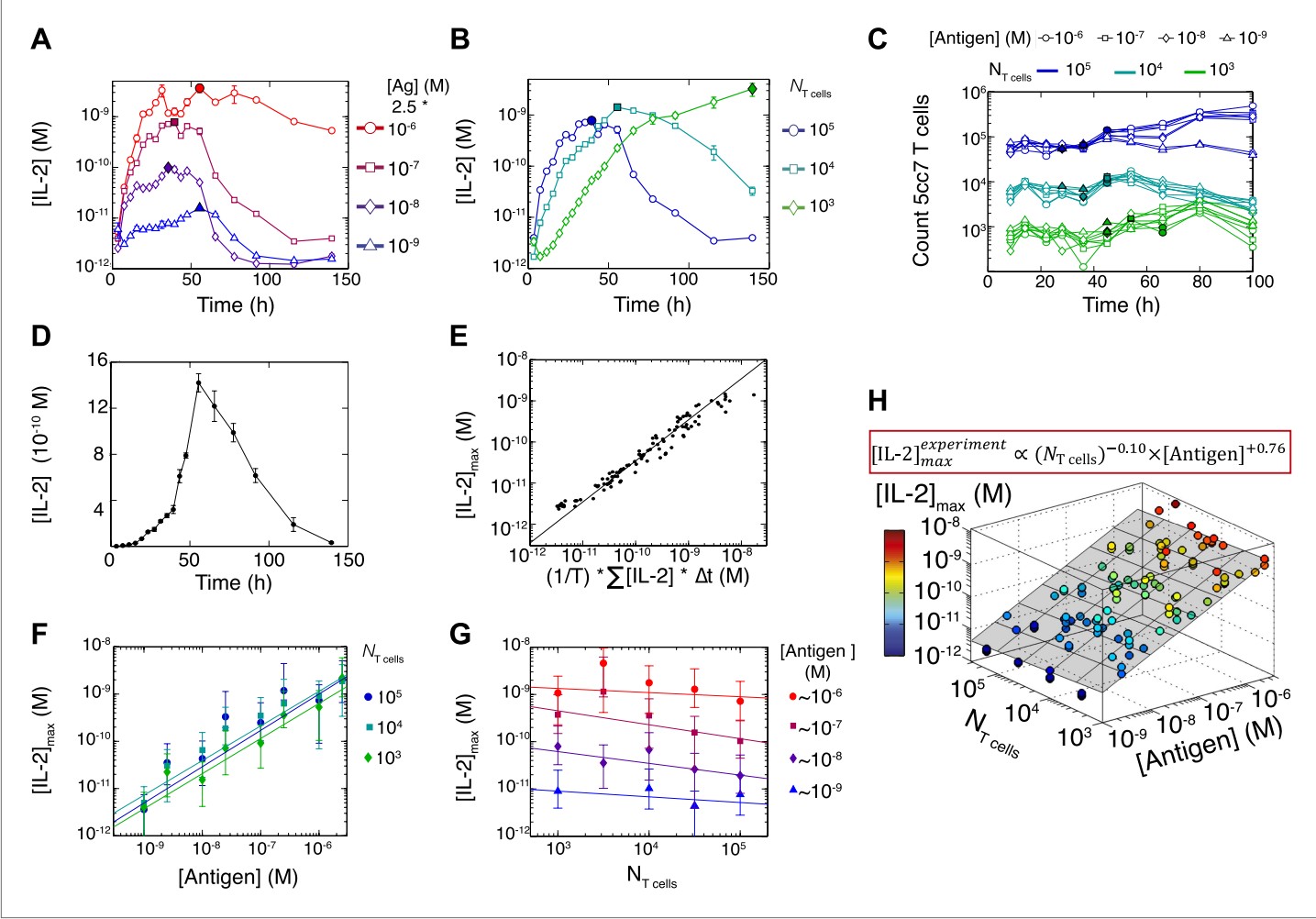

**Figure 2**. Dynamics and scaling of IL-2 production and consumption by T cells in vitro. For T cell cultures described in *Figure 1*, (**A** and **B**) supernatants were collected at different timepoints (typically every 4 hr), and [IL-2] was measured by ELISA. For each condition, we recorded the maximal concentration of accumulated [IL-2] ([IL-2]$_{max}$, filled symbol). Data are represented as mean ± SEM. (**C**) Number of live T cells in cultures as a function of time. For a given quantity of stimulating antigen (indicated by the symbol), the filled point marks the time at which the cytokine accumulation of smaller populations (10$^4$ and 10$^3$ T cells) is equal to or surpasses the larger population's [IL-2]$_{max}$ (10$^5$ cells). (**D**) Nonlinear accumulation and consumption of IL-2 for a culture of 10$^4$ T cells activated with 5.10$^5$ B10.A *Cd3e$^{-/-}$* splenocytes pre-pulsed with 250 nM of K5 antigen illustrated in linear scale. (**E**) Correlation of [IL-2]$_{max}$ with the total accumulated [IL-2] over time for 118 different conditions (varied doses of antigen and varied numbers of T cells) over six experiments. (**F**) [IL-2]$_{max}$ scales almost linearly with [Antigen] over a large dynamic range. (**G**) [IL-2]$_{max}$ is practically independent of $N_{Tcell}$. These data were compiled from independent experiments in which T cells were stimulated with either 1, 2, or 2.5 * 10$^{-6}$/10$^{-7}$/10$^{-8}$/10$^{-9}$ M antigen. Thus, results in (**G**) are grouped according to order of magnitude of antigen dose. (**H**) Scaling law for experimentally determined [IL-2]$_{max}$ as a function of [Antigen] and $N_{Tcell}$. The grey plane is fitted for the PLSR result.

results. IL-2 receptor exposure to only 10 pM of IL-2 is biophysically sufficient to trigger ligand binding (*Smith, 1988*), and subsequent STAT phosphorylation sharply inhibits IL-2 synthesis (*Long and Adler, 2006*; *Villarino et al., 2007*; *Waysbort et al., 2013*). Accordingly, T cells should be incapable of producing much more than 10 pM of cytokine (*Figure 3*). Empirically, however, IL-2 accumulation readily exceeded this concentration (*Figure 2*). In *Figure 3F* we show that the two-dimensional fit of the predicted scaling exponents for the classical model

$$[IL-2]^{predicted}_{max} \propto (N_{T\,cells})^{+0.25(\pm0.005)} \times [Antigen]^{+0.32(\pm0.008)}, \qquad (2)$$

was also incompatible with our experimental results (*Equation 1*). These contradictions prompted further investigation of the IL-2 pathway.

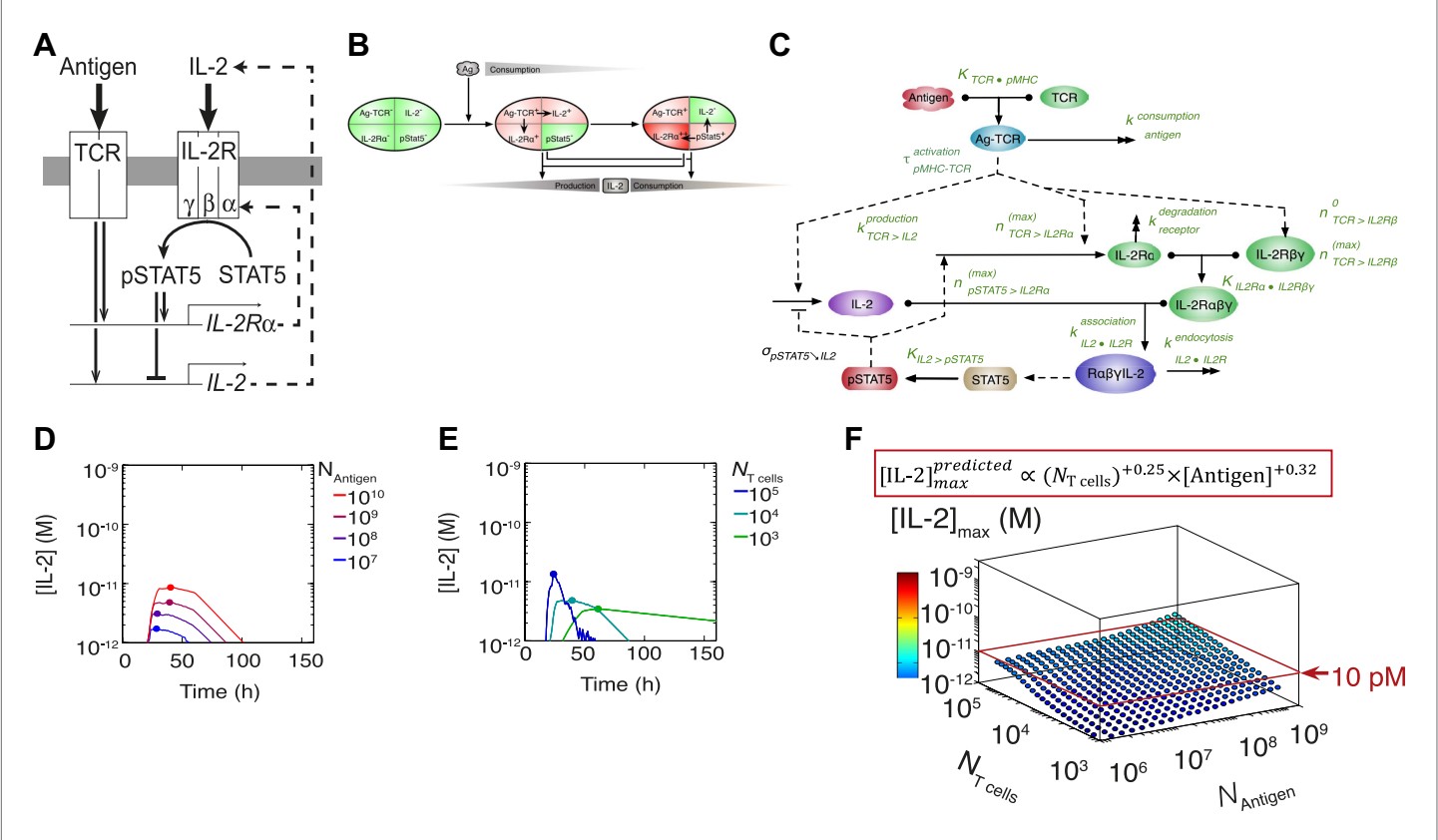

**Figure 3.** Shortcomings of the classical model of the IL-2 pathway. (**A**) Sketch of the classical pathway for IL-2 secretion and consumption. (**B**) Cartoon representation of progression through cellular states during production and consumption of IL-2: from naive (IL-2Rα⁻, IL-2⁻) to activated IL-2 producers (IL-2Rα⁺, IL-2⁺ and IL-2Rα⁺) and finally to IL-2 consumers (IL-2Rα⁺⁺, IL-2⁻). (**C**) Biochemical model of IL-2 regulation as described in the literature ('classical model'); parameters in green are derived from experiments. Classical model prediction of IL-2 dynamics for (**D**) $10^4$ T cells stimulated with varied quantities of antigen and (**E**) varied numbers of T cells stimulated with $10^8$ molecules of antigen. (**F**) Two-dimensional dependency for [IL-2]$_{max}$ as a function of [Antigen] and $N_{Tcell}$, as predicted by the above classical model. The red border represents the theoretical 10 pM ceiling of [IL-2] that cells can secrete before switching off IL-2 secretion.

## TCR inhibition of pSTAT5 tightly links IL-2 production to antigen signaling through a coherent feed-forward loop

To investigate this discrepancy, we systematically probed the dynamics of the IL-2 pathway by measuring IL-2 concentrations via ELISA, as well as cytokine receptor levels (IL-2Rα and IL-2Rβ) and response (pSTAT5) by flow cytometry. Visualizing the joint kinetics of cytokine, receptor and phospho-signal among activated cells, we found a striking antigen-dependency in the trajectory of the IL-2 pathway (*Figure 4A*, left). The pSTAT5 response was proportional to the product of IL-2Rα abundance and accumulated [IL-2] over time leading up to [IL-2]$_{max}$, but the efficiency of STAT5 phosphorylation lessened with increasing antigen dose (*Figure 4A*, right). This observation is consistent with previous reports of TCR signaling inhibiting the pSTAT5 response to IL-2 (*Lee et al., 1999*; *Yamane et al., 2005*).

We further characterized the effect of antigen dose on IL-2 signaling by sampling the responses of differentially activated T cells to titrated concentrations of exogenous IL-2 (*Figure 4B*). 48 hr after antigen activation, cells were collected and stripped of pre-bound cytokine with a low pH buffer, then washed and rested before exposure to serial dilutions of recombinant mouse IL-2. After 10 min of incubation with IL-2, cells were fixed, permeabilized and stained for pSTAT5, IL-2Rα, IL-2Rβ, and γc for flow cytometry analysis. Examining the geometric mean of pSTAT5 in activated (IL-2Rα⁺) T cells, we found that the quantity of stimulating antigen did not affect the average EC$_{50}$, i.e., sensitivity, of IL-2 response (*Figure 4B*). Instead, we observed that increasing antigen dose resulted in higher levels of IL-2 receptors at 48 hr (*Figure 4C*), yet paradoxically dampened the amplitude of STAT5

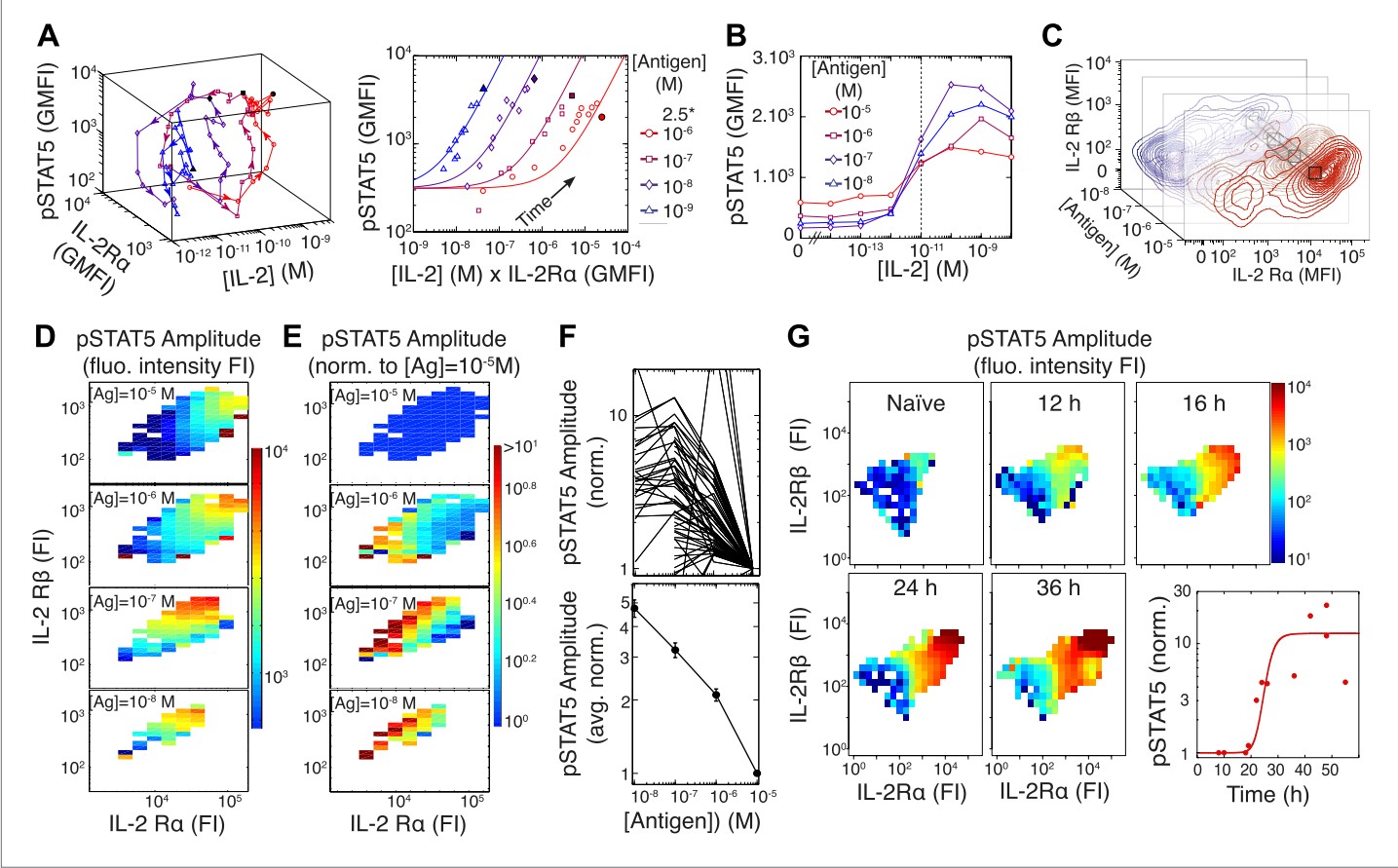

**Figure 4**. Experimental characterization of the antigen-driven inhibition of IL-2 signaling. (**A**) Dynamics of IL-2 pathway over 150 hr (arrows indicate progression in time) for $10^5$ 5C.C7 T cells activated in vitro by splenocytes pulsed with varied [K5] antigen (left). STAT5 phosphorylation was measured as the geometric mean fluorescence intensity (GMFI) at different times before reaching the maximal [IL-2] concentration (filled symbol), and correlated with the product of [IL-2] and IL-2Rα GMFI for activated cells (right, representative of more than four experiments). (**B**) STAT5 phosphorylation in response to exogenous IL-2 for cells 48 hr post activation with splenocytes pulsed with varied doses of antigen. pSTAT5 is reported as GMFI for all activated IL-2Rα⁺ T cells. (**C**) Distributions of the abundance of IL-2Rα and IL-2Rβ at 48 hr post activation with splenocytes pulsed with varied doses of antigen. Cell-to-cell variability analysis (CCVA) parses these distributions to compare the signaling responses among populations of cells (bins) defined by set levels of IL-2Rα and IL-2Rβ (e.g., black cross-section across antigen doses). (**D**) and (**E**) Cell-to-cell variability analysis, see Experimental Procedures for details. pSTAT5 responses for cultures in (**C**) were parsed according to binned levels of IL-2 receptors. Amplitudes of pSTAT5 for 10 nM ≤ [K5] ≤ 10 μM for each IL-2Rα/IL-2Rβ bin were presented (**D**) as fluorescence intensity (FI) or (**E**) as a FI normalized to the pSTAT5 amplitude for [K5] = 10 μM. (**F**) Normalized pSTAT5 amplitude are reported for individual bins of IL-2Rα and IL-2Rβ levels (top) or averaged across all IL-2Rα and IL-2Rβ levels (bottom). Error bars are computed as the SEM across all bins. (**G**) Cell-to-cell variability analysis of pSTAT5 response to IL-2 for varied levels of IL-2Rα and IL-2Rβ at different time points. Inset: time dependence of the average pSTAT5 amplitude measured for individual bins of IL-2Rα and IL-2Rβ over time (n = 3 independent experiments).

phosphorylation. Consequently, we postulated that antigen signaling inhibits IL-2 response not at the level of IL-2 receptor engagement, but rather at the level of STAT5 phosphorylation.

To disambiguate the antigen/IL-2 signaling cross-talk from concomitant changes in receptor abundance, we performed cell-to-cell variability analysis (CCVA) (*Cotari et al., 2013*) of IL-2 responsiveness (*Figure 4D–F*). Using our custom-designed flow cytometry analysis software, *ScatterSlice*, we parsed the populations of activated T cells into subpopulations (bins) of equal IL-2 receptor abundance, and calculated the dose response amplitude of pSTAT5 (color) within each bin (*Figure 4D*). The hindered IL-2 responsiveness of T cells activated with higher antigen doses was easily visualized through these heat maps. We then quantified the pSTAT5-inhibiting effects of TCR signaling by normalizing the pSTAT5 amplitude within each bin to the response of cells that were stimulated with the highest antigen dose (10 μM), yet expressed equivalent levels of IL-2 receptor α, β (*Figure 4E–F*) and γc chains (our unpublished data). By factoring out the dynamic variation in IL-2 receptor abundance associated

with T cell activation, CCVA demonstrated that TCR-driven inhibition of IL-2 signaling scales linearly with antigen dose (*Figure 4F*). Furthermore, CCVA showed that this inhibitory effect decays throughout the course of the T cell response (*Figure 4G*), independently of changes in IL-2 receptor levels. Our single cell analyses allowed the deconvolution of downstream signaling events from receptor abundance, and demonstrated the tunability of the inhibitory cross-talk between antigen and pSTAT5 signaling.

We hypothesized that this titrated antigen-driven inhibition of IL-2 signaling could delay pSTAT5-mediated shutdown of IL-2 production, especially in strongly activated cells, thus enabling the accumulation of IL-2 beyond the canonical pSTAT5 signaling threshold of 10 pM (*Smith, 1988*). We tested this intricate regulation of IL-2 production through signal blocking experiments, using a cytokine-capture assay to identify IL-2-secreting cells (*Figure 5*). We demonstrated that persistent TCR signaling is required to sustain IL-2 production (*Huppa et al., 2003*) at all times. Administration of an antibody that disrupted TCR-pMHC contact quickly shut down IL-2 production (*Figure 5A–B*) and concomitantly increased pSTAT5 within the population (*Figure 5C–D*), further suggesting inhibition of IL-2 production by pSTAT5 response, and of pSTAT5 by TCR signaling. Consistent with reports of negative feedback inhibition of IL-2 production by STAT5-mediated IL-2 signaling (*Long and Adler, 2006*; *Villarino et al., 2007*; *Waysbort et al., 2013*), blocking pSTAT5 via a chemical inhibitor of Janus kinase (JAK) activity increased the number of IL-2 producing cells (*Figure 5D*). Strikingly, in contrast to the rapid drop ($\tau_{drop} = 0.5 \pm 0.1$ hr) observed in antigen-blocked conditions, dual inhibition of IL-2 and TCR signaling resulted in a slower decline in IL-2 producers ($\tau_{drop} = 2.2 \pm 0.1$ hr) (*Figure 5D*). This demonstrates that pSTAT5 signaling following antigen withdrawal functions as a swifter mechanism to shut down IL-2 secretion compared to the loss of TCR signal alone.

This biochemical network (*Figure 5E*) forms a coherent feed forward loop, in which a signal (TCR) and its effect on a target (inhibition of pSTAT5) regulate a common output in the same direction (promoting IL-2 production). More specifically, this is a type IV coherent feed-forward loop, where one arm directly promotes an output, and the other arm represses an output's inhibitor (*Alon, 2007*). Theoretical

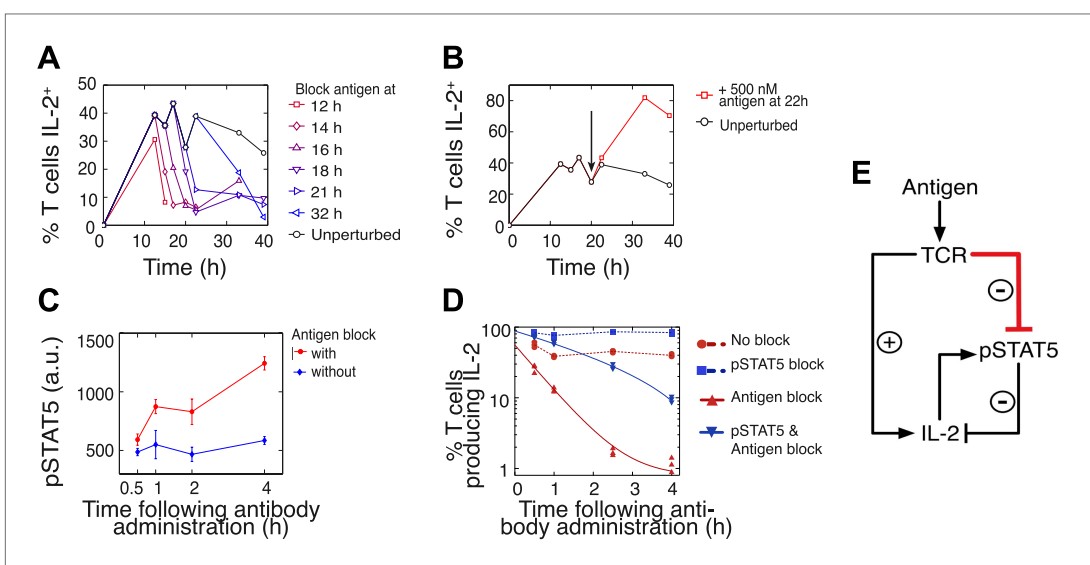

**Figure 5**. Coherent feed-forward loop regulation of IL-2 secretion. (**A** and **B**) $3 \times 10^4$ 5C.C7 TCR-transgenic *Rag2*$^{-/-}$ T cells co-cultured with $3.5 \times 10^5$ APCs pre-pulsed with 500 nM K5 antigen. (**A**) Blocking of cognate pMHC ligand via administration of 20 µg/ml α-I-E$^k$ antibody at varied time points during IL-2 production causes a rapid drop (detected here within 2 hr) in the number of IL-2-producing cells. (**B**) Addition of K5 antigen at t = 22 hr to cultures increases the numbers of IL-2-producing cells. (**C**) Phosphorylation of STAT5 is rapidly enhanced upon blocking of cognate pMHC ligand via administration of 20 µg/ml α-I-E$^k$ antibody. (**D**) Cells activated with 1 µM of K5 antigen were treated with a JAK inhibitor or carrier control at 9 hr. Cells of each condition were then treated with antigen-blocking reagent anti-I-E$^k$ or control anti-H2-D$^b$ at 30 hr. IL-2 production was measured via cytokine capture assay. All conditions performed in triplicate. (**E**) Sketch of the antigen-driven inhibition of IL-2 signaling (red), which makes IL-2 production contingent on antigen availability.

studies of such motifs have highlighted their signal delay properties and mutual exclusion of target and repressor signals (**Mangan and Alon, 2003**), as has been observed for IL-2 and pSTAT5 (**Long and Adler, 2006**). Our experiments additionally revealed that this regulatory architecture enforces tight synchronization between output production and sustenance of input cues. Through antigenic control of pSTAT5-mediated repression, IL-2 synthesis is neither terminated before nor extended beyond the loss of antigen signaling.

## Accelerating rate of IL-2 production per cell generates population size-independence and expands dynamic range of IL-2 accumulation

In parallel, we quantified IL-2 secretion for varied T cell population sizes. Using the cytokine-capture assay, we found that greater fractions of smaller populations of T cells maintained IL-2 secretion for longer periods of time (**Figure 6A**), consistently with in vivo studies (**Sojka et al., 2004**). Paradoxically, though smaller populations of T cells ($10^3$) could accumulate greater concentrations of IL-2 ([IL-2]$_{max}$) than larger populations (**Figure 2B,G**), they yielded far fewer IL-2-producing cells (**Figure 6B**). To 'catch up' in IL-2 accumulation without converging in numbers of IL-2 producers, T cell population size must adjust cellular rates of IL-2 secretion and/or consumption.

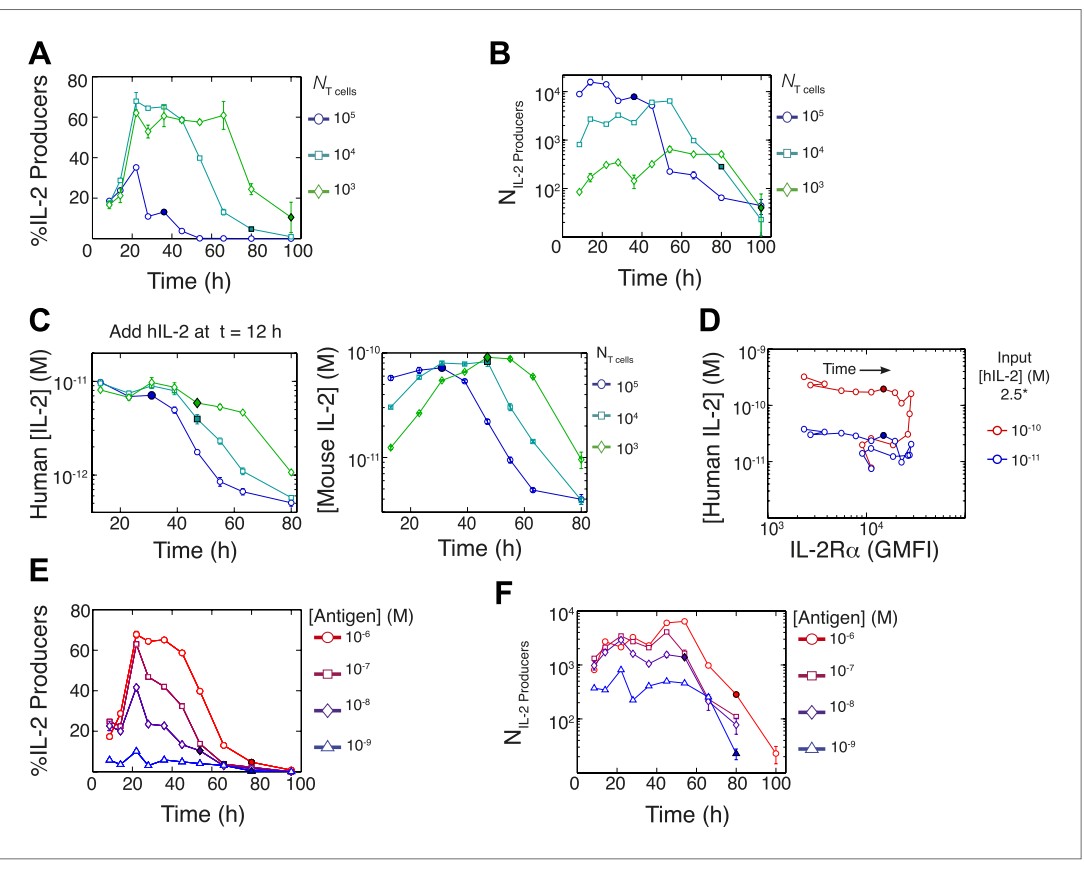

**Figure 6**. Count of IL-2 producing cells and rate of IL-2 consumption do not account for scaling law in IL-2 accumulation. We quantitate IL-2 production at the individual cell level for cultures as described in **Figure 1**. Percentage (**A**) and counts (**B**) of T cells producing IL-2 as a function of time for varied numbers of 5C.C7 T cells, activated with splenocytes pulsed with 1 μM K5 antigen. (**C**) Negligible IL-2 consumption during IL-2 production phase. Varied numbers of 5C.C7 T cells were activated by splenocytes pulsed with 1 μM K5 antigen and cultured with 10 pM of human IL-2 added 12 hr post initial activation, in triplicate. Left: human IL-2 and Right: mouse IL-2 detected in cultures over time. Graphs are representative of three experiments. (**D**) Depletion of added human IL-2 as a function of IL-2Rα upregulation. 50,000 5C.C7 T cells were stimulated with 200,000 APCs pulsed with 500 nM K5 antigen during experiment tracking the consumption of 250 or 50 pM human IL-2 added at 6 hr after the start of co-culture. Timepoints were taken every 6 hr between 6 and 96 hr of culture. Percentage (**E**) and counts (**F**) of T cells producing IL-2 as a function of time for $10^4$ T cells activated with splenocytes pulsed with varied doses of K5 antigen. DOI: 10.7554/eLife.01944.008

We characterized the effect of population size on IL-2 consumption by adding human IL-2 to large and small populations of T cells 12 hr after antigen activation (*Figure 6C*). Both mouse and human IL-2 (hIL-2) bind equivalently to the IL-2 receptor but can be measured separately by ELISA (*Deenick et al., 2003*), allowing us to resolve consumption from production. We found that both large and small populations showed similarly limited consumption of hIL-2 before their respective times of maximal mouse IL-2 accumulation (*Figure 6C*). The onset of hIL-2 consumption correlated with high pSTAT5-driven upregulation of IL-2Rα, which reached its apex several hours after cells amassed maximal mouse IL-2 (*Figure 6D*). Therefore, the observed parity in IL-2 accumulation between differently sized T cell populations cannot be attributed to differential cytokine consumption during the IL-2 secretion period.

In previous studies, single-cell measurements established that T cells are bimodal in secreting IL-2 at early (<6 hr) timepoints; stronger antigenic stimuli increases the number of IL-2-producing cells, but not the amount of IL-2 produced per cell (*Podtschaske et al., 2007*; *Huang et al., 2013*). Our time series experiments did confirm that greater antigenic stimulus resulted in larger numbers of IL-2 secreting cells over several days (*Figure 6E–F*). However, the antigen scaling of the number of IL-2 producers was insufficient to account for the observed power law in accumulated IL-2 (*Figure 2F,H*). We reasoned that if bimodality in IL-2 production indeed sets a constant IL-2 secretion rate per cell (*Podtschaske et al., 2007*), the concentration of cytokine should increase linearly with the cumulative number of secreting cells over time. Surprisingly, we observed the emergence of a nonlinear relationship between these two quantities (*Figure 7A–B*), demonstrating a time-dependent acceleration in the rate of IL-2 production.

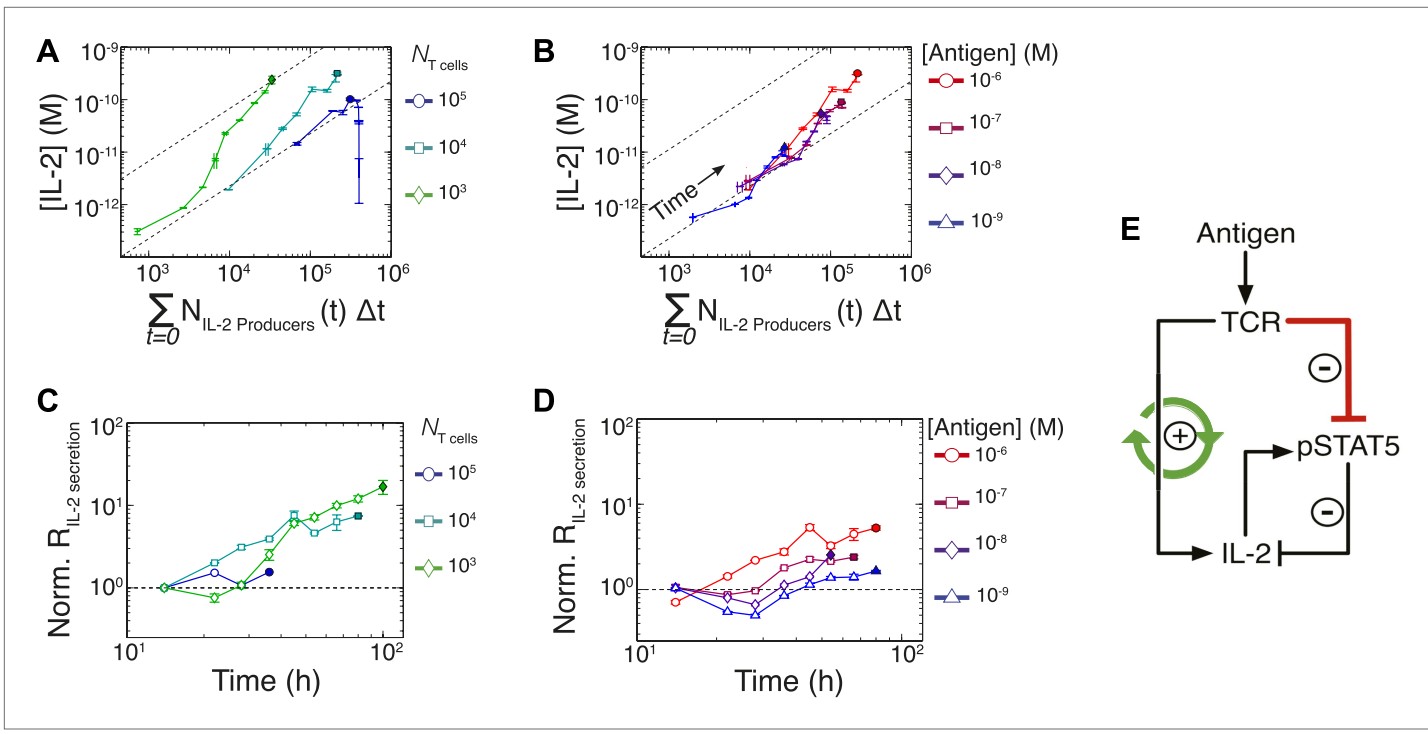

**Figure 7**. Experimental characterization of the nonlinear acceleration of IL-2 secretion in activated T cells. (**A** and **B**) The integral of the number of IL-2 producers over time following activation is compared to [IL-2] (measured in duplicate) accumulated (**A**) for 1 μM K5 antigen with different numbers of T cells or (**B**) for $10^4$ T cells exposed to different quantities of antigen. (**C** and **D**) Fold increase from the average initial rate of IL-2 production per cell as a function of time for varied number of T cells (**C**) or for varied doses of antigens (**D**). The average rate of IL-2 secretion was estimated to be 7.5 molecules per second (*Figure 8—figure supplement 1*). (**E**) Sketch of the positive feed-forward loop (in green) accounting for the acceleration in IL-2 secretion over sustained periods of antigenic stimulation.

The following figure supplements are available for figure 7:

**Figure supplement 1**. Criticality of IL-2 boost to achieve antigen-scaling and population-size-independence in [IL-2]$_{max}$.

Despite different stimulating conditions, at early timepoints (<20 hr) activated T cells secreted IL-2 at similar rates (*Figure 7C–D*, dashed lines & *Figure 8—figure supplement 1*). However, as smaller populations of T cells sustain longer periods of IL-2 production (*Figure 6A*), they demonstrated greater time-dependent increases in their rate of IL-2 accumulation per cell (up to 30-fold) (*Figure 7C*). Ultimately, this acceleration allows them to surpass the IL-2 maxima of 100-fold larger populations in a short amount of time (*Figure 2C*), despite having fewer IL-2 producers (*Figure 6B*). In parallel, T cells stimulated with low doses of antigen maintained a constant secretion rate for longer periods of time, slowly building their acceleration in cytokine production, while strongly-activated T cells increased their apparent secretion rate sooner (*Figure 7D*). This time- and antigen-dependent acceleration in the rate of IL-2 secretion (*Figure 7E*) amplified the effect of small differences in the duration and amplitude of T cell signaling (Supplement to *Figure 7*). It explained the observed nonlinear kinetics of IL-2 accumulation, and underlied the population size-independence of $[IL-2]_{max}$ and its expanded dynamic range.

## Computational model of the IL-2 pathway demonstrates the significance of experimentally characterized feedbacks

To understand how these newly found regulations contribute to the emergence of the observed IL-2 scaling laws, we employed biochemically explicit computational modeling of the IL-2 pathway to build an ordinary differential equation model (see 'Materials and methods') that captured the experimentally observed progression of molecular states within individual T cells (*Figure 8A*). Through TCR and subsequent IL-2 signaling, respectively, cells advance from a naïve state to a state of cytokine production, and ultimately to a state of pure cytokine consumption. The amount of antigen per cell regulates IL-2 production through the experimentally characterized inhibitory feed-forward (*Figure 5*) and stimulatory feedback (*Figure 7*) loops. The experimentally parameterized theoretical implementation of these regulatory elements is diagrammed in *Figure 8B* and explained in detail in 'Materials and methods–Full Model Implementation'.

TCR-mediated inhibition of pSTAT5 signaling is modeled as a reduction in the catalytic ability of the IL2/IL-2R complex (IL-2Rαβγ•IL-2) to induce STAT5 phosphorylation by a factor proportional to the amount of antigen-engaged TCR (Ag-TCR). This mechanism captures the experimental observation that TCR crosstalk modulates the amplitude, but not the $EC_{50}$, of IL-2 response (*Figure 4B*).

In modeling the time-dependent acceleration in IL-2 secretion, we followed several lines of evidence that suggested that this feedback depends on antigen signaling. First, this acceleration could be observed despite perturbation of JAK, Phosphoinositide 3-kinase (PI3K) and CD28 activity (our unpublished data). Moreover, greater amounts of available antigen and lower numbers of T cells yielded the largest accelerations in IL-2 production (*Figure 7C–D*); these conditions are known to increase the length of T cell interactions with antigen presenting cells (*Garcia et al., 2007*). Indeed, recent studies have shown that the duration of antigen priming signals strongly impacts gene expression in T cells (*Tubo et al., 2013*), particularly the upregulation of IL-2 (*Henrickson et al., 2013*). Furthermore, antigen-experienced cells have been shown to exhibit higher rates of IL-2 secretion per cell (*Huang et al., 2013*), possibly through TCR-driven epigenetic modification of the IL-2 locus (*Bruniquel and Schwartz, 2003*). We therefore postulated that strength and persistence in TCR signaling determines the extent of acceleration in IL-2 secretion. To model this, we introduced a phenomenological variable, *Boost*, which upon activation (*Boost$_a$*) increases the rate of IL-2 production per cell. We parameterized *Boost*'s initial activation by TCR signals to be slow, such that sustained TCR engagement was required to substantially accumulate *Boost$_a$*. Activated *Boost* then catalyzes further *Boost$_a$*, generating a positive feedback that results in the non-linear dynamics of IL-2 secretion. Such phenomenological feedback recapitulates the observed time-dependent acceleration in IL-2 secretion, which is most potent for high quantities of antigen and low numbers of T cells (*Figure 7*).

Since antigen and secreted IL-2 are shared by the whole T cell population, the number of T cells determines the amount of antigen and cytokine available per cell in the model. Thus, T cell population size regulates the global rate of IL-2 accumulation by setting the number of producers and their antigen availability over time. Additionally, population size controls the global rate of IL-2 depletion by determining the number of consumers, and by dynamically regulating their IL-2 depletion capabilities: the persistent availability of antigen to smaller T cell populations delays pSTAT5-mediated upregulation of IL-2Rα, which postpones the initiation of IL-2 consumption (*Figure 6C–D & 8C*). While accurately predicting IL-2 consumption will require accounting for cell proliferation and death, which exert stronger effects on longer (>3 day) timescales (*Figure 2C*), our model reproduces the measured

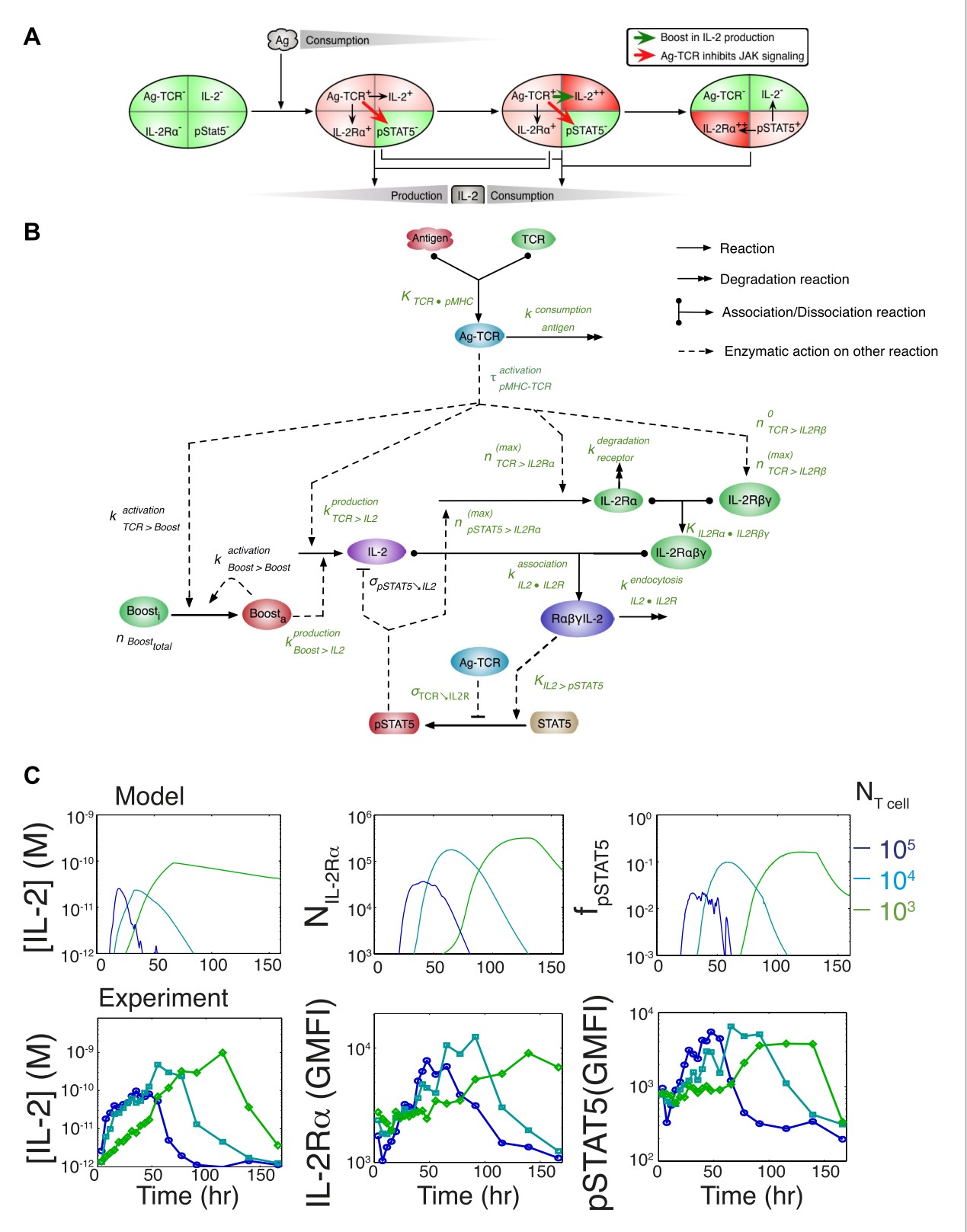

**Figure 8**. Computational model of IL-2 pathway. (**A**) Cartoon representation of progression through cellular states during production and consumption of IL-2. Highlighted arrows indicate new regulation uncovered in **Figures 5 and 7**. (**B**) Molecular reaction network of IL-2 pathway used to build the mathematical model on the basis of **Figure 8A**. Detailed description of the model is given in 'Materials and methods'. Chemical reactions are

*Figure 8. Continued on next page*

*Figure 8. Continued*

represented by solid lines, and dashed lines represent the enzymatic activity of chemical reactions. The parameters associated with the reactions are indicated in the diagram and the values of the parameters are listed in the 'Materials and methods'. Experimentally determined/estimated parameters are colored in green. Phenomenologically determined parameters are colored in black. (**C**) Comparison of model-simulated (top row) and experimentally observed (bottom row) temporal dynamics of IL-2, IL-2Rα and pSTAT5 for three different numbers of 5C.C7 T cells in 200 µl medium. T cells are co-cultured with $5.10^5$ APCs prepulsed with 25 nM of K5 antigen; in the model simulation, the antigen dose is $1 \times 10^8$ molecules. Kinetics are representative of six independent experiments.

The following figure supplements are available for figure 8:

**Figure supplement 1**. Additional experimental measurements to parametrize the computational model.

dynamics of the IL-2 production pathway for different quantities of antigens and numbers of T cells (*Figures 8C and 9A*). Most significantly, it recapitulates the scaling law (*Figure 9A* bottom):

$$\left[IL-2\right]_{max}^{model} \propto \left(N_{T\,cells}\right)^{-0.12(\pm0.03)} \times \left(N_{Antigen}\right)^{+0.82(\pm0.02)}. \tag{3}$$

We then used our model to examine the relevance of the regulatory mechanisms uncovered in *Figures 5 and 7*. In *Figure 9B*, we tested the importance of the TCR-mediated inhibition of IL-2 signaling. Our complete model accounts for the antigen-dependency of pSTAT5 gain with increasing levels of IL-2Rα and IL-2 (*Figures 4A and 9B*, top). Removing this inhibitory crosstalk eliminates the antigen dose dependency of pSTAT5 response (*Figure 9B*, middle). It also results in a decreased antigen-dependency of [IL-2]$_{max}$ that flattens at ~10 pM (*Figure 9B*, bottom), the classical threshold for STAT5 phosphorylation (*Wang and Smith, 1987*), and thus for termination of IL-2 production (*Long and Adler, 2006*). Therefore, TCR inhibition of IL-2 signaling is critical to sustain cytokine secretion beyond 10 pM of IL-2.

We also explored the significance of the nonlinear acceleration in IL-2 production per cell observed in our experiments. The complete model captures the nonlinear correlation of IL-2 with cumulative numbers of IL-2 producing cells (*Figures 7A and 9C*, top). In contrast, abrogating the acceleration in IL-2 production yields a simple linear correspondence between these variables (*Figure 9C*, middle), and prevents small populations of T cells from accumulating comparable amounts of IL-2 to those secreted by larger populations. The lack of acceleration also decreases the scaling exponent of [IL-2]$_{max}$ with antigen dose (*Figure 9C*, bottom), as the antigen-dependence of IL-2 accumulation is limited to differences in the cumulative number of IL-2 producing cells (Supplement to *Figure 7*). Therefore, our experimentally determined, quantitative model illustrates the criticality of these new regulatory elements, through which T cells achieve population size-independent power law antigen scaling of IL-2.

We then tested our computational model of IL-2 pathway regulation through *in silico* and in vitro perturbation of STAT5 signaling. We blocked IL-2 signaling in the model by setting the STAT phosphorylation rate to zero. Our model predicted over ten-fold greater IL-2 accumulation in pSTAT5-inhibited vs unperturbed conditions (*Figure 9D*, top). Moreover, it forecasted that larger populations of T cells would sustain higher concentrations of IL-2 than smaller populations (*Figure 9D*, top left). Experimentally treating cells with a JAK inhibitor at time 0 confirmed these predictions, and validated our model's projections for the dynamics of IL-2 accumulation following JAK blockade (*Figure 9D*, bottom). These computational and experimental results demonstrate that the empirical scaling of IL-2 accumulation is critically dependent on feedbacks from IL-2 signaling.

## Model predicts the maintenance of IL-2 scaling and the inter-clone titration of TCR-pSTAT5 cross-talk in a two-clone setting

To further probe the functional significance of our model of IL-2 scaling, we tested numerically and experimentally the joint IL-2 response of two TCR transgenic T cell clones co-cultured at different densities and stimulated with varying concentrations of their respective cognate antigens. The model predicted and experiments confirmed that [IL-2]$_{max}$ for a mixed population of T cell clones is determined by the combined antigen doses, independently of cell numbers (*Figure 10A*; *Figure 10—figure supplement 1A–C*). This result demonstrates that IL-2 is a collective measure of global antigenic load with the potential to coordinate polyclonal responses.

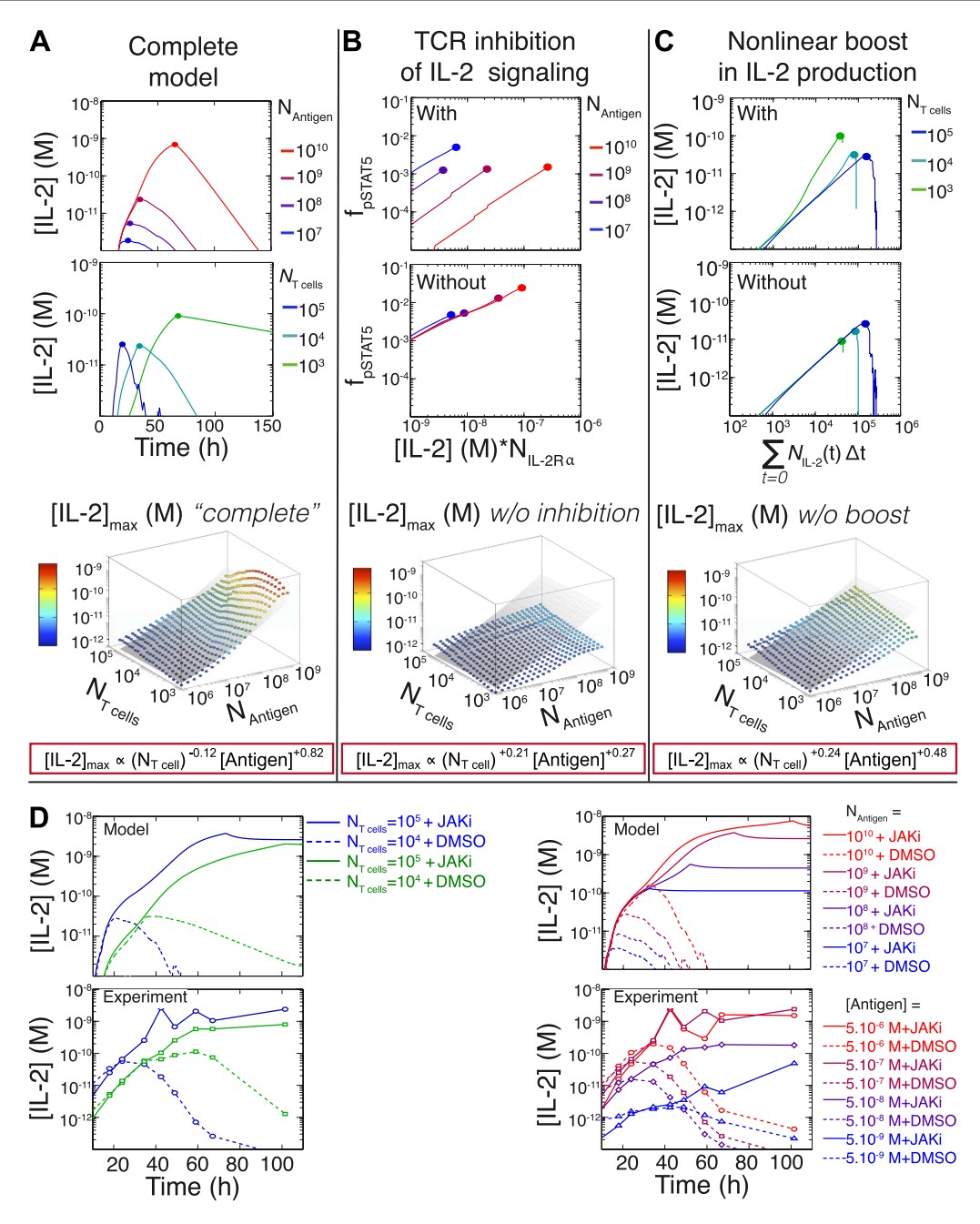

**Figure 9**. Mathematical modeling accounts for the scaling law in IL-2 dynamics. (**A**) Simulated dynamics of [IL-2] for different quantities of antigen molecules, $N_{Antigen}$ (top) and numbers of T cells, $N_{T\,cell}$ (middle). [IL-2]$_{max}$ dependency with $N_{Antigen}$ and $N_{T\,cell}$ (bottom-filled circles) can be fitted with **Equation 3** (bottom–grey plane). (**B**) Dependence of pSTAT5 response on $N_{Antigen}$, with (top) or without (middle) TCR inhibition of pSTAT5. Abrogating TCR inhibition leads to low saturation in [IL-2]$_{max}$ and spurious scaling with $N_{T\,cell}$ and $N_{Antigen}$ (bottom). (**C**) Our model recaptures the acceleration of IL-2 secretion as a function of cumulative numbers of IL-2 producing cells (top). Upon removing the boost in IL-2 secretion (middle), low $N_{T\,cell}$ fail to accumulate comparable [IL-2]$_{max}$ to high $N_{T\,cell}$ (bottom). (**D**) Model prediction (top row) and experimental validation (bottom row) of IL-2 accumulation kinetics with JAK inhibitor (JAKi—solid line) or without (DMSO—dashed line) for different numbers of T cells (left) activated with different quantities of K5 peptide (right).

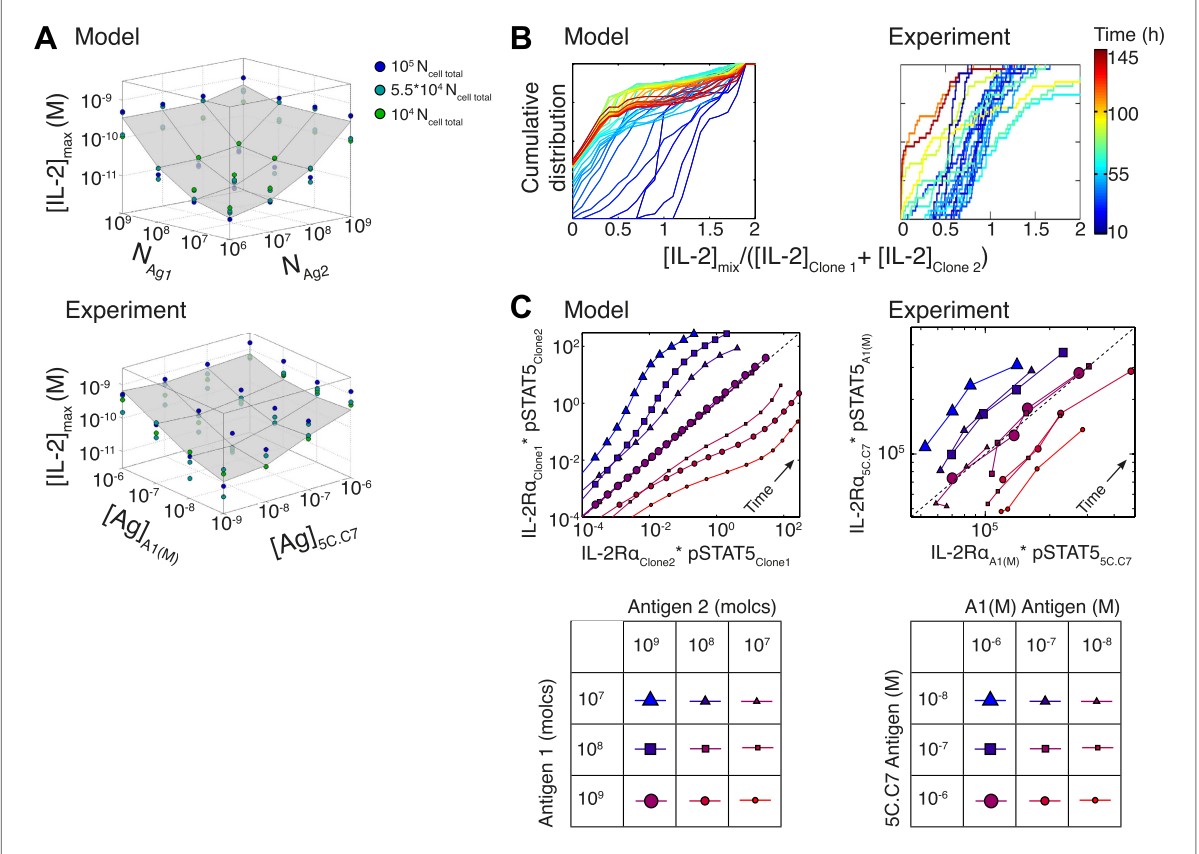

**Figure 10**. Testing the model of IL-2 regulation through mixed culture of two T cell clones. 5C.C7 and A1(M) TCR transgenic T cells were cultured at varied precursor frequencies ($5.10^3$ and $5.10^4$ T cells/well) with titrated concentrations ($10^{-6}/10^{-7}/10^{-8}$ M) of cognate antigens (K5 and HY peptides) pulsed on separate splenocytes. Graphs show two experiments and are representative of three experiments. (**A**) Model predictions (top) and experimental validation (bottom) of $[IL-2]_{max}$ for mixed cultures. Color represents total number of T cells ($5.10^3 + 5.10^3$, $5.10^3 + 5.10^4$, $5.10^4 + 5.10^4$). (**B**) Model (left) and experimental (right) cumulative distributions of the ratio of [IL-2] accumulated by the mixed culture to the sum of [IL-2] accumulated independently by each clone over all conditions. (**C**) Model prediction (left) and experimental validation (right) that pSTAT5 response to shared IL-2 can resolve the relative activating doses of antigen for $5.10^4$ cells of each clone. Marker shape: [Antigen1] = [K5], marker size: [Antigen2] = [HY], color: ratio [K5]/[HY].

The following figure supplements are available for figure 10:

**Figure supplement 1**. Additional computational predictions and experimental validation for a mixed culture of two T cell clones.

We compared the IL-2 concentration from mixed cultures to the sum of the IL-2 accumulated independently by each clone by quantifying the ratio of these two values (*Figure 10B*). For each time point (colored lines), the $[IL-2]_{mix}/([IL-2_{Clone1}] + [IL-2_{Clone2}])$ ratios were represented as a cumulative distribution of all 36 culture conditions. Throughout the IL-2 production phase (t < 50 hr, blue lines), the distributions were centered around a ratio of 1, indicating that IL-2 produced by 2 clones in the same well is approximately equal to the sum of the IL-2 made by the same two populations in separate wells. This result suggests that the IL-2 production phase is dictated by a T cell population's TCR stimulation, regardless of other nearby immune reactions. At later timepoints (red lines in *Figure 10B*), the cumulative distribution of ratios of [IL-2] in co-cultures to the sum of the IL-2 from individual clones was drastically shifted to the left, as less IL-2 remained in the co-culture wells due to the increased numbers of IL-2 consumers. This was observed most strikingly when mixing a large number of poorly activated T cells with a small number of strongly activated T cells, as predicted by our model.

Furthermore, the model and experiments both demonstrated that each clone's relative antigen dose could be resolved through the shared cytokine environment via proportional inhibition of the IL-2 pathway (*Figure 10C*). STAT5 phosphorylation in single cells is determined by the global concentration of IL-2 and cells' individual IL-2 responsiveness—a convolution of cellular receptor abundance and

antigen signaling (**Figure 4A**). Simple algebra ('Materials and methods–Algebraic relationship between pSTAT5, IL-2Rα and Antigen for a mixture of two T cell clones) demonstrates how the cue-signal-response to IL-2 in a mixed population of cells predicts the distribution of stimulating antigens:

$$\frac{(\#\,pSTAT5)_2 * (\#\,IL-2R\alpha)_1}{(\#\,pSTAT5)_1 * (\#\,IL-2R\alpha)_2} = \frac{f([Antigen_2])}{f([Antigen_1])}.$$

Remarkably, this theoretical expression was confirmed experimentally. The relative doses of antigen signals for 5C.C7 and A1(M) T cells, which are of similarly high quality, were detectable through their individual responses to shared IL-2 (**Figure 10C**). Accordingly, the signaling crosstalk unraveled in **Figure 4** allows a mixed population of T cells to perform both a local (antigen) and a global (cytokine) measurement of collective stimulus through the IL-2 pathway. Furthermore, TCR cross-talk inhibition of the IL-2 pathway provides a graded readout of antigen signaling within activated cells with quantitative resolution across a polyclonal system. We thus sought to apply the potent sensing capacity of this cross-talk toward the quantification of tissue antigenicity.

## IL-2 pathway trajectories provide a sensitive measurement of tumor antigenicity

To illustrate how the TCR-IL-2R crosstalk can be used to quantify antigen availability in vivo over a large dynamic range, we probed T cell responses against cells harvested from explanted melanoma tumors and their draining lymph nodes. (**Figure 11A**). B6 mice were injected in the right flank with $10^5$ B16 melanoma cells, then sacrificed two weeks post-injection. Their tumors and tumor-draining lymph nodes were harvested for co-culture with naïve TRP-1 transgenic T cells, which are specific for the TRP-1 melanoma antigen (**Muranski et al., 2008**). As the tumors varied proportionally in area, weight and cellularity, we asked if the TCR responses of TRP-1 cells would scale with tumor size by re-suspending all tumors in the same volume. In parallel, cells from tumors and tumor-draining lymph nodes were re-suspended at a uniform concentration to probe for differences in antigen presentation per cell. To calibrate this cross-talk assay, spleen and lymph node cells from non tumor-bearing B6 mice were pulsed with titrated amounts of TRP-1 peptide. Throughout the first 50 hr of in vitro activation, we measured the concentration of IL-2 and abundance of IL-2Rα and pSTAT5 for activated TRP-1 T cells in all co-cultures. We then assessed the antigen-induced inhibition of pSTAT5 by calculating the rate of gain in pSTAT5 with increasing IL-2 and IL-2Rα ($Slope_{Xtalk}$) (**Figure 11B**, left). As in **Figure 4A**, we found inverse proportionality between $Slope_{Xtalk}$ and titrated concentrations of antigenic peptide, which established a calibration curve to back-calculate fold changes in the antigenic capacity of tumor tissue suspensions (**Figure 11B**, right).

We found that the calculated amount of antigen signaling experienced by TRP-1 T cells scaled with tumor cellularity (**Figure 11B**, right): larger tumors, which yielded more input cells per normalized volume, induced greater suppression of IL-2 signaling, and were thus estimated to have a proportionally greater antigen load (inset). Concurrently, when tumor cells were suspended at the same cellular concentration, the induced $Slope_{Xtalk}$ were very similar, indicating equivalent quantities of presented antigen per cell (**Figure 11C**, left). Normalizing the antigen calculated for each tissue source by the number of cells plated, we confirmed that both tumor dilution strategies yielded very similar estimates of antigen presentation per cell, which did not depend on tumor size (**Figure 11D**, green and red). As expected, measurements of $Slope_{Xtalk}$ estimated that tumor-draining lymph node cells presented much less antigen than tumor suspensions (**Figure 11C**, right , **Figure 11D**, blue). Strikingly, our measurements showed that the antigen presented per cell in the draining lymph nodes did scale with tumor size (**Figure 11D**). Our method thus confirms anticipated biological phenomena: tumors with 10 times more cells have 10 times more antigen, there is equal cellular antigenicity between clonal tumors, and there is significantly less antigen per cell in lymph nodes vs tumors.

In sum, scalable inhibition of the IL-2 pathway can resolve a wide dynamic range of antigen quantities. This inhibitory cross-talk underlies the $[IL-2]_{max}$ scaling presented in **Figure 2**, but can be captured with far fewer timepoints; in fact, a single snapshot measurement of the amplitude of pSTAT5 response to high dose IL-2 can distinguish relative strengths of TCR signaling among activated T cells (**Figure 4**). These measurements may help quantify differences in TCR signal strength between conditions resulting in equal binary T cell activation but discrepancies in downstream effector function of activated cells (**van Heijst et al., 2009**), a problem that is particularly acute in the field of tumor immunology (**Joncker et al., 2006**; **Engelhardt et al., 2012**).

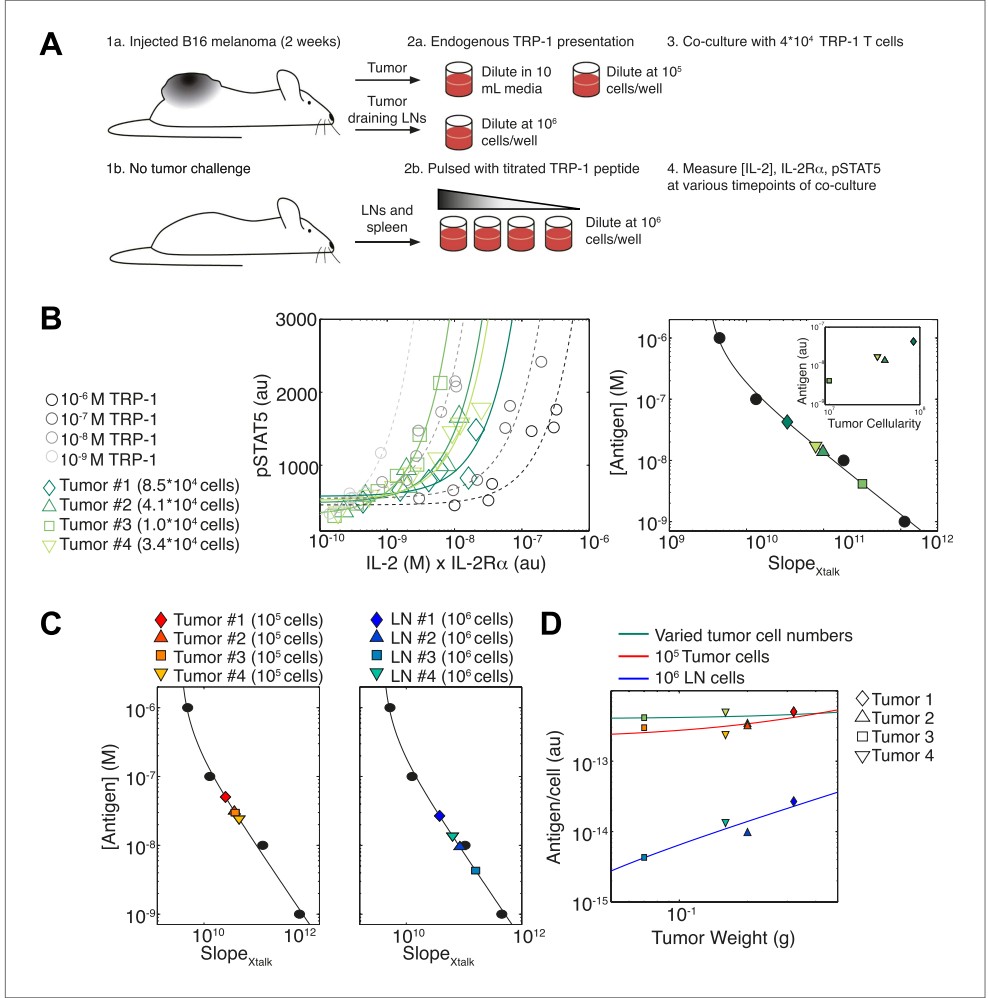

**Figure 11**. Applying antigen-driven inhibition of IL-2 signaling to estimate tumor antigenicity. (**A**) Schematic of experimental design. (**B**) Left: pSTAT5 increase over the first 52 hr of culture as a function of IL-2 and IL-2Rα for titrated TRP1 peptide pulsed on C57BL/6 splenocytes (Calibration Series, gray) and tumor samples diluted in equivalent volumes (Sample Series, green). Trajectories were fit with the equation $pSTAT5(t) = Slope_{Xtalk} \cdot ([IL-2](t).\#IL-2R\alpha(t) + Background)$. Graphs are representative of four experiments. Right: antigen dose as a function of $Slope_{Xtalk}$, as established by Calibration Series (black). Calibration curve allows estimation of effective antigenicity of tumor samples. Antigenicity of tumors scales with tumor cellularity (insert). (**C**) Back-calculated antigenicity for $10^5$ cells of each mouse's tumor (left) and $10^6$ cells of each mouse's tumor draining lymph nodes (right). (**D**) Correlation of tumor weight to estimations of antigen presentation per cell for each tissue. DOI: 10.7554/eLife.01944.016

## Discussion

Our study revealed a scaling law for IL-2 accumulation as a function of antigen dose, and demonstrated that the collective reporting of antigen load through IL-2 production is independent of the number of T cells in mono- and poly-clonal populations. It also quantitatively dissected the regulatory architecture required for this cell–cell communication of antigen input through shared cytokine output. We found that the inhibitory cross-talk between antigen and IL-2 signaling generated a coherent feed-forward loop architecture that ensured direct correspondence between the persistence of antigenic cues and the duration of cytokine production. In parallel, we found a time-dependent acceleration in the rate of IL-2 secretion that allows small T cell populations to compensate for having 20-fold fewer IL-2 producers (*Figure 6B*) in less than twice as much time (*Figure 2B,C*), and establishes better dynamic range between different doses of antigen. Hence, time integration of IL-2 regulatory loops generates collective-level outputs that reflect global antigen abundance with

higher fidelity, functional range and temporal persistence than early activation responses (*Figure 1*) (*Cheong et al., 2011*).

This feedback-controlled, titrated accumulation of IL-2 may help scale and direct the long-term responses of activated cells according their degree of antigenic stimulus. IL-2 signaling critically optimizes lymphocyte differentiation, proliferation, and survival (*Williams et al., 2006*; *Bachmann et al., 2007*; *Pipkin et al., 2010*; *Liao et al., 2011*; *McNally et al., 2011*; *Boyman and Sprent, 2012*). In the case of in vivo CD8⁺ T cell differentiation, sustained IL-2-driven IL-2Rα expression corresponds to the adoption of short-lived effector, rather than memory, fates (*Kalia et al., 2010*; *Obar et al., 2010*; *Pipkin et al., 2010*). The tight correlation we find between antigen load and IL-2 accumulation may ensure that large pathogenic challenges are communicated through large IL-2 availability to generate robust effector responses. Indeed, another in vivo study has shown that the expression of IL-2Rα at day four post-infection scales with antigenic affinity, and correlates to greater effector cell expansion and survival (*Zehn et al., 2009*; *Kalia et al., 2010*). The higher IL-2Rα levels on OT-I cells exposed to bacteria expressing high affinity OVA vs less potent Q4 peptides suggest that T cell responses to higher affinity epitopes trigger greater accumulation of IL-2 in vivo. Accordingly, we have observed in vitro scaling of $[IL-2]_{max}$ and pSTAT5 inhibition with antigen quality across this altered peptide ligands series (KT, GV, GA-B, unpublished data). Moreover, recent quantitative studies relating CD4⁺ (*Tubo et al., 2013*) and CD8⁺ (*Henrickson et al., 2013*) T cell differentiation to the abundance and persistence of TCR priming signals in vivo have implicated the antigen-scaled production and sensing of IL-2 as possible mediators of divergent cell fate decisions. Finally, graded inhibition of pSTAT5 by TCR signaling may directly influence CD4⁺ T helper subtype differentiation by blocking time-sensitive cytokine signals (*Yamane and Paul, 2012*).

This study also introduces the possibility of detecting analog tuning within digitally activated populations by examining cross-talk pathways. Positive and negative feedback loops within proximal TCR signaling generate a sharp digital activation filter that is necessary for efficient discrimination of antigenic ligands (*Altan-Bonnet and Germain, 2005*; *Das et al., 2009*). As a result, many of the readouts downstream of TCR activation (i.e., CD69, CD25, ppERK–see *Figure 1A*) are all-or-none, and gradations in signal strength are resolved only by counting the number of activated T cells. Such 'percent activated' measurements have been fruitfully used to survey tissue antigenicity (*Badovinac and Harty, 2002*; *Nandi et al., 2009*), but binary single parameter readouts carry some limitations: they are prone to saturation of input detection range, exhibit narrow output dynamic ranges, are subject to shifts in value by cellular death and migration, and provide little information on the functional capacity of activated cells. In contrast, quantifying the degree of TCR-driven inhibition of pSTAT5 signaling inside activated T cells provides an indication of the strength of signaling within the responding population. While these measurements are not immune to saturation, convolution of antigen-dependent STAT5 phosphorylation efficiency with its consequent scaling of IL-2 production (*Figures 4A, 10C, 11*) increases output resolution of the initiating antigen load. Thus, the regulatory architecture uncovered in this study, which expands the scaling of cytokine production to antigen input, can be similarly exploited to make more informative measurements of antigen signaling (*Figure 11*).

Finally, our study demonstrated how reductionist, systems biology approaches can quantify the shape, strength and kinetics of immunological regulation when the molecular mechanisms are unknown. By observing shifts in multi-dimensional trajectories of the IL-2 pathway as a function of input numbers of antigens and T cells, we deduced the presence of previously unaccounted for regulatory interactions. Our simplified experimental system allows detailed probing of endogenous pathways with high-resolution time- and dose response- series that strongly constrain our mathematical model, and our quantitative understanding, to the observed biology. Through this bottom-up approach (*O'Garra et al., 2011*), one can dissect the dynamic regulation and scaling laws of cytokine communications that underlie complex in vivo settings.

## Materials and methods

### Media, reagents and antibodies

#### Media

All in vitro experiments were performed in complemented RPMI medium (prepared by the Media facility at MSKCC), which consists of RPMI 1640 supplemented with 10% heat–inactivated fetal bovine serum, 2 mM L–glutamine, 10 mM HEPES (pH 7.4), 0.1 mM non–essential amino acids, 1 mM sodium

pyruvate, 100 µg/ml of penicillin, 100 µg/ml of streptomycin and 50 µMβ–mercaptoethanol. All cell cultures were maintained in an incubator at 37°C with 5% CO2.

## Reagents and antibodies

K5 peptide (sequence: ANERADLIAYFKAATKF) was obtained from GenScript (Piscataway, NJ). HY peptide (sequence: REEALHQFRSGRKPI) was obtained from Peptide 2.0 (Chantilly, VA). TRP-1 peptide (sequence: SGHNCGTCRPGWRGAACNQKILTVR) was obtained from Genemed Synthesis (San Antonio, TX).

The following antibodies were obtained from eBioscience (San Diego, CA):

- anti–IL–2 capturing antibody (Clone: JES6–1A12)
- biotinilated anti–IL–2 detection antibody (Clone: JES6–5H4)
- anti–IL–2Rγ (anti-CD132, Clone: 4G3 – fluorescent tag: PE)

The following antibodies were obtained from BD Bioscience (San Jose, CA);

- anti–CD4 (Clone: RM4–5—fluorescent tag: Alexa700, Pacific Blue)
- anti–IL–2Rα (anti–CD25, Clone: PC61 – fluorescent tag: PE–Cy7, PerCPCy5.5)
- anti–IL–2Rβ (anti–CD122, Clone: TM–β1– fluorescent tag: PE)
- anti–Vb3 TCR (Clone: KJ25– fluorescent tag: PE)
- anti–Vb8.1–8.2 TCR (Clone: KJ16– fluorescent tag: FITC)
- anti-Vb14 TCR (Clone: 14-2—fluorescent tag: FITC)
- anti-CD45.1 (Clone: A20—fluorescent tag: FITC)

The monoclonal antibody against phospho–STAT5(Tyr 694) (Clone: C11C5) was obtained from Cell Signaling (Danvers, MA).

The polyclonal antibody against Rabbit IgG (fluorescent tag: APC) was obtained from Jackson ImmunoResearch (West Grove, PA);

The blocking antibody, anti–MHC class II molecule I–E$^k$ (clone 14–4–4S; American Type Culture Collection) and control antibody, anti–MHC class I molecule H2–D$^b$ (clone 20–8–4S; American Type Culture Collection) were produced by Memorial Sloan Kettering Cancer Center (MSKCC) Monoclonal Antibody Core Facility (MACF).

Mouse IL–2 Secretion Assay–Detection Kit (PE) was obtained from Miltenyi Biotec (Auburn, CA). Recombinant mouse IL–2 (mIL–2) was obtained from eBioscience (San Diego, CA).

4',6–diamidino–2–phenylindole (DAPI) was obtained from Sigma–Aldrich (St. Louis, MO).

Cell Trace Violet (CTV) and Live/Dead Fixable Yellow (LDY) were obtained from Molecular Probes (Eugene, OR),

The chemical inhibitor of JAK1/JAK2 was AZD1480, a generous gift from Dr Jackie Bromberg (MSKCC).

## Mice and cell culture

Experiments used lymph node and spleen cells from 5C.C7 *Rag2$^{-/-}$*, A1(M) *Rag1$^{-/-}$* or TRP-1 *Rag1$^{-/-}$* TCR–transgenic mice and B10.A *Cd3e$^{-/-}$* or C57BL/6 mice aged 2–6 months, cultured in comple-mented RPMI. B10.A *Cd3e$^{-/-}$* or C57BL/6 splenocytes pulsed overnight with varying concentrations of K5, HY or TRP-1 peptide were co–cultured with varying numbers of T cells in 200 µl of media in flat–bottomed 96–well plates. The Institutional Animal Care and Use Committee of Memorial Sloan Kettering Cancer Center approved all of the animal experiments.

## Time series and perturbations

At each timepoint, supernatants were collected and stored at −20°C. Cells were collected and fixed in 1.6% cold PFA on ice, permeabilized with 90% MeOH, and stored at −20°C. T cell numbers were quantitatively assessed by flow cytometry: diluted, CTV-labeled samples from each well were stained without any spin steps and run on a BD LSRII for 60 s at a calibrated flow rate, allowing back-calculation of the absolute number of CD45.1 5C.C7 T cells. IL-2 producer counts $N_{IL–2^+}$ were obtained by multi-plying the above 5C.C7 T cell counts by the frequency of IL-2 producing cells in the same well, meas-ured via the IL–2 Secretion Assay Kit (Miltenyi Biotech). After completion of the time series, levels of ppERK (*Altan-Bonnet and Germain, 2005*), pSTAT5 and IL-2 receptor components were measured by FACS, and supernatant [IL-2] was measured by ELISA (see *Feinerman et al. 2010* for details of the

method). The integral of the number of IL-2 producers over time was calculated as $\sum_{i=0}^{i_T} N_{IL-2^+}(i) \ast (t_{i+1} - t_i)$, where $i_T$ is the index for the last measurement at $t = T$.

Antigen-blocking experiments were performed with anti–MHC class II molecule I–E$^k$ (clone 14–4–4S) or anti–MHC class I molecule H2-D$^b$ (clone 20–8–4S) as a control antibody, at a final concentration of 20 µg/ml. Inhibition of pSTAT5 signaling was performed using Jak inhibitor AZD1480 at a final concentration of 10 µM.

## Correlating the total amount of secreted IL-2 with [IL-2]$_{max}$

Total amount of secreted IL-2 was calculated as the integral of [IL-2](t) over time t, using the method of trapezoidal approximation, normalized over the duration of the experiment and reported in the molar unit M. T$_{final}$ ranges from 80 to 150 hr depending on the experimental conditions. We found a linear correlation between the total amount of secreted IL-2 with [IL-2]$_{max}$ ($\chi^2$ = 4.3 for 118 points and 1 parameter):

$$[IL-2]_{max} = 0.34 \ast \left( \frac{1}{T_{final}} \int_0^{T_{final}} dt\, [IL-2](t) \right)$$

## Cell-to-Cell Variability Analysis of IL-2 responsiveness

Cells were stripped of pre-bound cytokine by 2 min incubation in ice–cold pH 4.0 0.1 M glycine buffer, then washed 2X in RPMI and rested for 10 min at 37°C (*Feinerman et al., 2010*). Cells were added to cytokine titrations and incubated for 10 min at 37°C before fixation, permeabilization and FACS analysis. Single, CD4$^+$ IL–2Rα$^+$ cells were then identified using FlowJo and further analyzed using our custom–designed processing R program *ScatterSlice* (*Cotari et al., 2013*) (this software can be downloaded at www.Scatterslice.org). Each sample's CD4$^+$ IL–2Rα$^+$ cells were divided into bins according to varied levels of cytokine receptors. Within each bin, a three parameter Hill equation fit the pSTAT5 base, amplitude and EC$_{50}$ for the IL-2 dose titration. For each stimulation condition, fitted amplitudes for different levels of receptors were then normalized relative to the pSTAT5 amplitude in the corresponding bin within the T cell population stimulated with the highest dose of antigen (10 µM K5). For each antigen dose, the mean and standard error of the mean of normalized pSTAT5 amplitude was calculated across all occupied bins of IL–2Rα, β, γ expression.

## Error estimation of exponents in [IL-2]$_{max}$ scaling law

[IL–2]$_{max}$ was fit across a range of doses of antigen ([Antigen]) and numbers of T cells ($N_{T\,cell}$) by partial least squares regression (PLSR) for each independent experiment to obtain exponents for [Antigen] and $N_{T\,cell}$. The average and standard error of mean of the exponents was calculated from $n_{experiment}$ = 6 individual fits. Error bars correspond to the 95% confidence interval.

## Mathematical model

We constructed a chemical reaction network (*Figure 8B*) to model IL-2 production and signaling in a population of T cells based on measurements from previous literature and our experiments. Our model comprises 2 global variables: the numbers of free antigen (Ag) and IL-2. These molecules are shared by all cells in the medium. For each cell, there are five independent state variables (where • represents a complex) representing the number of molecules of: Ag, IL-2Rα, IL-2Rβγ, IL-2Rαβγ•IL-2 (where IL-2Rαβγ is the full IL-2 receptor, i.e., IL-2Rα•IL-2Rαβγ and Activated Boost, and 3 dependent variables (calculated from the independent variables): Ag•TCR, IL-2Rαβγ and Fraction pSTAT5. The total numbers of TCR and Boost remain constant over time. We observed that the fraction of IL-2 producing cells increases linearly with time: this was modeled phenomenologically by introducing a uniformly distributed random delay (between 10 and 60 hr) in the first encounter between T cell and antigen.

We implemented a bottom-up approach to simulate different numbers of T cells ($N_{T\,cell}$). We simulated $n_{cell}$ number of cells (typically $n_{cell}$ = 20), and scaled the association/dissociation of Ag to TCR, production of IL-2, and the association/dissociation of IL-2 to the IL-2Rα by a factor $N_{T\,cell}/n_{cell}$ to capture the dynamics of $N_{T\,cell}$ number of T cells in $V$ = 200 µl culture medium. We generated a set of nonlinear ordinary differential equations (see below) describing the dynamics of the variables for $n_{cell}$ number of cells. We solved these sets of stiff nonlinear ordinary differential equations using the MATLAB CVODE solver (*Hindmarsh et al., 2005*). See below for details of model.

## Full model implementation

We constructed a chemical reaction network to model IL-2 production and signaling in a population of T cells. Our experimental system tracked how large populations of T cells processed macroscopic numbers of molecules of antigen, IL-2, IL-2 receptors and signaling molecules. Moreover, the topology of our network does not include strong positive feedbacks (*Vilar et al., 2002*; *Artomov et al., 2010*; *Francois et al., 2013*), where stochasticity in chemical reactions yielded qualitatively different output. Hence, we relied on a deterministic framework to build an ordinary differential equation model. This model is based on previous literature describing IL-2 signaling and contains the previously determined components (*Feinerman et al., 2010*; *Cotari et al., 2013*):

- T cells constitutively express IL-2Rβ and γ (CD122 and CD132).
- Engagement of TCR by antigen leads to activation of T cells.
- Activated T cells upregulate IL-2Rβ, express the surface receptor IL-2Rα(CD25) and secrete the cytokine IL-2.
- The IL-2 receptor chains pre-form a heterotrimeric complex, IL2Rαβγ.
- Secreted IL-2 accumulates in the extracellular medium and binds to T cells' IL-2 receptor components to assemble a full tetrameric complex, IL-2/IL2Rαβγ.
- The full complex phosphorylates STAT5 and is endocytosed and degraded (allowing IL-2 consumption).
- Phosphorylated STAT5 (pSTAT5) enacts both a negative and positive feedback on its own signaling by shutting down IL-2 production and upregulating production of IL-2Rα, respectively.

This classical model of IL-2 cue-signal-response (*Figure 3*) is insufficient to generate the experimentally observed scaling law (*Equation 1*; *Figure 2H*). We appended two new regulatory elements as experimentally characterized in *Figures 5 and 7*:

- Our quantitative measurements of the IL-2 signaling pathway demonstrate that TCR signaling inhibits the phosphorylation of STAT5. This TCR-mediated inhibition of IL-2 signaling occurs in an antigen dose-dependent manner.
- Single cell measurements of the number of IL-2-producing cells indicate that the rate of IL-2 production per cell accelerates nonlinearly with time.

The full reaction network of the model is given in *Figure 8B* and the nonlinear ordinary differential equations of the model are given in 'Equations'. We obtained quantitative parameters from the literature, or parameterized the associated chemical reactions based on our own measurements as detailed below ('Parameters'). As our previous study validated a well-mixed approximation to model our experimental conditions (*Feinerman et al., 2010*), we did not include spatial considerations for the cell-to-cell communications via extracellular IL-2 in these in vitro settings. Our complete model is reductionist by nature and fully parameterized: it enables us to account for our experimental observations (*Figure 9*), and to make predictions that we validated experimentally (*Figures 9 and 10*).

## TCR-pMHC interaction

Engagement of antigen to TCR is modeled by a simple chemical equilibrium upon interaction between T cell and antigen-presenting cells. We used the equilibrium constant from recent in situ measurements for I-E$^k$/MCC antigen and 5C.C7 TCR (*Huang et al., 2013*). Interactions of I-E$^k$/K5 antigen with 5C.C7 TCR was measured in vitro (*Corse et al., 2010*), and found to be similar to I-E$^k$ interactions with MCC antigen. Hence we took

$$K^{equilibrium}_{TCR \cdot IE^k / K5} = 30,000.$$

As previously demonstrated (*Kedl et al., 2002*; *Schulz et al., 2009*), the loss of antigen signaling is a critical parameter of the long-term response of T cells. We implemented an exponential temporal loss of antigen from the surfaces of antigen-presenting cells based on engagement with TCR. The typical timescale for such process has been measured in *von Essen et al. (2004)*:

$$k^{consumption}_{antigen} = \frac{\ln(2)}{3.5} h^{-1}.$$

## IL-2R expression and IL-2 secretion

### IL-2R subunit expression

We modeled the TCR signal-dependent production of IL2-Rα as all-or-none according to a threshold of TCR-antigen engagement, and limited to a low plateau of IL-2Rα (*Sheldon et al., 1993*). The linear pSTAT5-mediated upregulation of IL-2Rα is modeled by multiplying the fraction of phosphorylated STAT5 per cell by the maximum mode of IL-2Rα per cell (*Cotari et al., 2013*). The rates for reaching the two modes of IL-2Rα synthesis are expressed as their respective equilibrium plateaus multiplied by the published degradation rate of IL-2Rα (*Duprez et al., 1988*).

IL-2Rβ is limiting compared to IL-2Rγ (*Cotari et al., 2013*), so we modeled its kinetics to account for the abundance of IL-2 receptor signaling chains. IL-2Rβ is expressed at a basal levels of 1000 copies in unactivated cells (*Cotari et al., 2013*) and undergoes a ten-fold increase in abundance following antigen engagement above a digital threshold (*Figure 8—figure supplement 1*).

### Nonlinear acceleration in IL-2 production

IL-2 secretion is also bimodal (*Huang et al., 2013*). We therefore modeled IL-2 production as an all-or-none response to signaling input, with Ag-TCR and pSTAT5 acting as activating and repressing factors (*Long and Adler, 2006*; *Villarino et al., 2007*), respectively ('Equations'). The number of Ag-TCR sufficient to trigger IL-2 production has been measured to be 1 (*Huang et al., 2013*). The initial rate of IL-2 production is set at 7.5 molecules per cell per second based on our measurements (*Figure 8—figure supplement 1*).

In order to incorporate the experimentally observed nonlinear acceleration in IL-2 production per cell (*Figure 7*), we introduced a phenomenological variable, 'Boost', an activatable cellular mediator that increases the IL-2 secretion rate per cell. Though the exact molecular nature of this acceleration remains elusive, we have constructed our model to capture its experimentally documented quantitative characteristics (*Figure 7*): IL-2 production per cell increases up to thirty times the basal rate, in an antigen signaling-dependent manner. In our model, we establish that TCR engagement activates Boost from its inactive state (Boost$_i$) to its active state (Boost$_a$). To incorporate nonlinear acceleration, we allowed Boost$_a$ to autocatalytically activate the remaining Boost$_i$. The fraction of activated Boost per cell is then multiplied by the maximal Boost-dependent rate of IL-2 production. As a result, so long as IL-2 production is sustained, that is TCR is engaged with antigen (*Figure 5A*), the production of IL-2 per cell is a nonlinearly increasing function of time. We parameterized the dynamics of Boost and the production rates of IL-2 such that the model captures the experimentally observed correlation of [IL-2] with cumulative numbers of IL-2 producing cells (*Figure 7*).

## IL-2 binding and signaling

IL-2 receptor chains pre-form heterotrimeric complexes, even in the absence of IL-2 (*Cotari et al., 2013*). As in our previous study (*Feinerman et al., 2010*), we treated IL-2Rβ and IL-2Rγ as a single component IL-2Rβ/γ, which then binds to the IL-2Rα subunit. This association at the membrane is very fast relative to the timescale of protein synthesis; we therefore modeled the pre-association of the IL-2 receptor chains as a steady state process for each time step (*Cotari et al., 2013*). The number of IL-2Rαβγ complexes at each time step is thus calculated as the root of a quadratic equilibrium function of the abundances of IL-2Rα and β/γ and the affinity constant for their binding (*Cotari et al., 2013*).

IL-2 then binds to the heterotrimeric IL-2Rαβγ forming the full complex IL-2Rαβγ•IL-2 with a fast association rate (*Wang and Smith, 1987*; *Smith, 1988*; *Rickert et al., 2004*). The unoccupied IL-2Rα and IL-2Rβ/γ degrade at very slow rates, with half-lives of 7 hr and 39 hr, respectively (*Duprez et al., 1988*). In contrast, the full complex IL-2Rαβγ•IL-2 is rapidly internalized by T cells, with a half-life of 15 min (*Hemar et al., 1995*).

We modeled the phosphorylation of STAT5 as a sigmoid dose response function of the full complex, IL-2Rαβγ•IL-2. As IL-2Rαβγ•IL-2 formation is limited by the abundance of IL-2Rβ/γ, the relationship between IL-2Rαβγ•IL-2 and STAT5 phosphorylation does not saturate and thus remains linear, as observed in *Figures 4A, 10C and 11B*. TCR-driven inhibition of IL-2 signaling (*Figure 4*) was modeled by allowing Ag-TCR to inhibit the catalytic activity of the full complex, IL-2Rαβγ•IL-2. The strength of the inhibition was parameterized to reproduce (*Figure 9B*) our experimental observations given in *Figure 4*.

## Testing the effect of JAK inhibition on IL-2 accumulation

The first test of our model consisted of modeling the effect of Janus kinase (JAK) inhibition on the accumulation of IL-2. This model prediction was made by setting the levels of STAT5 phosphorylation to 0 (hence abrogating the positive feedback on IL-2Rα and the negative feedback on IL-2 secretion). The results of these simulations were compared to the experimental validation in *Figure 9D*.

## Modeling a mixture of two different T cell clones

We then tested our model with a simulation of a mixture of two distinct T cell clones co-cultured at different population densities, and activated by varying doses of their respective cognate antigens. As in the single clone model, we used the 'bottom up' approach ('Mathematical model') to model of a mixture of two clones. For each clone we simulate $n_{cell}$ = 20. We varied ($N_{Tcell}^{Clone1}/n_{cell1}$) and ($N_{Tcell}^{Clone2}/n_{cell2}$) to simulate various numbers of T cells for clone 1 and clone 2, respectively. There are three global states: $Ag_1$ (antigen activating clone 1), $Ag_2$ (antigen activating clone 2), and [IL-2], which is shared by all cells from both clones. For each cell, there are five independent state variables representing the number of molecules of: Ag, IL-2Rα, IL-2Rβγ, IL-2Rαβγ•IL-2, and Activated Boost, and three dependent variables (calculated from the independent variables): Ag•TCR, IL-2Rαβγ and Fraction pSTAT5. Comparison of the two clone model predictions and their experimental validation are presented in *Figure 10*.

## Equations

We modeled $\mathcal{N}_{cells}$ undergoing activation in a volume $\mathcal{V}$. Our previous work (*Feinerman et al., 2010*) validated the well-mixed approximation to model IL-2 communications over long timescales. To limit integration times, we modeled $n_{cell}$ individual cells within a volume $\mathcal{V}.n_{cells}/\mathcal{N}_{cells}$. To reproduce the linear increase of the number of activated T cell over time, we assumed that each individual cell (labeled $i$) within this $n_{cell}$ cohort gets activated at a time $t_i$ (typically, $n_{cells}$ = 50). These $t_i$ represent the time when the $_i{}^{th}$ T cell encounters an antigen-presenting cell. Hence, at a given time $t$, the number of activated cells $n_{activated}(t)$ is:

$$n_{activated}(t) = \sum_{i=1}^{n_{cell}} \mathcal{H}(t, t_i),$$

where $\{t_i\}_{i=1\ldots n_{cell}}$ are the times of activation of the $n_{cell}$ cells being simulated, and $H(x, y)$ is the Heaviside (thresholding) function, defined as:

$$\mathcal{H}(x, y) = \begin{cases} 0 & : & \text{for } x < y \\ 1 & : & \text{for } x \geq y \end{cases}$$

Three variables are described with a steady-state approximation. They are the number of engaged TCR on cell $i$ ($n_{TCR\bullet pMHC_i}$), the number of preformed receptors for IL-2 on cell $i$ ($n_{IL2R_i}$) and the amount of STAT5 phosphorylation within cell $i$ ($\mathcal{P}_{pSTAT5_i}$) associated with engagement of $IL2R_i$ with IL-2. In all equations, $n_X$ represents the absolute number of molecule $X$.

We compute the amount of complexes between pMHC and TCR, or between the α and β chains of the IL-2 receptor. For a thermodynamic equilibrium between $X$ and $Y$,

$$X + Y \rightleftharpoons X \bullet Y,$$

the amount of complex $\mathcal{C}$ for $X \bullet Y$ is a function $n_X^0$, $n_Y^0$ and the equilibrium constant $K_{equilibrium}$:

$$\mathcal{C}\left(n_X^0, n_Y^0, K_{equilibrium}\right) = \frac{1}{2}\left(n_X^0 + n_Y^0 + K_{equilibrium} - \sqrt{\left(n_X^0 + n_Y^0 + K_{equilibrium}\right)^2 - 4.n_X^0.n_Y^0}\right).$$

$$n_{TCR\bullet pMHC_i}(t) = \begin{cases} 0 & : & \text{for } t < t_i \\ \mathcal{C}\left(n_{TCR}^0, n_{pMHC}(t)/n_{activated}(t), K_{TCR\bullet pMHC}\right) & : & \text{for } t \geq t_i \end{cases}$$

$$n_{IL2R_i}(t) = \mathcal{C}\left(n_{IL2R\alpha_i}(t), n_{IL2R\beta\gamma_i}(t), K_{IL2R\alpha\bullet IL2R\beta}\right)$$

$$\mathcal{P}_{pSTAT5_i}(t) = \frac{n_{IL2 \bullet IL2R}(t)}{K_{IL2>pSTAT5} + n_{IL2 \bullet IL2R}(t)} \times \frac{1}{1 + \sigma_{TCR \searrow IL2R} . n_{TCR \bullet pMHC_i}(t)}$$

The ordinary differential equations for the other variables are:

$$\frac{dn_{pMHC}(t)}{dt} = -k_{antigen}^{consumption} \left( \sum_{i=1}^{n_{cell}} n_{TCR \bullet pMHC_i}(t) \right)$$

$$\frac{dn_{IL2R\alpha_i}(t)}{dt} = k_{receptor}^{degradation} . \left[ n_{TCR>IL2R\alpha}^{(max)} . \mathcal{H}\left(n_{TCR \bullet pMHC_i}(t), \tau_{receptor}^{production}\right) \right. $$
$$\left. + n_{pSTAT5>IL2R\alpha}^{(max)} . \mathcal{P}_{pSTAT5_i}(t) - n_{IL2R\alpha}(t) \right]$$

$$\frac{dn_{IL2R\beta_i}(t)}{dt} = k_{receptor}^{degradation} . \left[ n_{IL2R\beta}^{(0)} + \left(n_{IL2R\beta}^{(max)} - n_{IL2R\beta}^{(0)}\right) . \mathcal{H}\left(n_{TCR \bullet pMHC_i}(t), \tau_{receptor}^{production}\right) \right]$$
$$- k_{receptor}^{degradation} . n_{IL2R\beta_i}(t)$$

$$\frac{dn_{IL2 \bullet IL2R_i}(t)}{dt} = k_{IL2 \bullet IL2R}^{association} . \frac{N_{cells}}{n_{cells} \mathcal{V} \mathcal{N}_a} \left[ n_{IL2R_i}(t) - n_{IL2 \bullet IL2R_i}(t) \right] . n_{IL2}(t)$$
$$- k_{IL2 \bullet IL2R}^{endocytosis} . n_{IL2 \bullet IL2R_i}(t)$$

$$\frac{dn_{Boost_i}(t)}{dt} = \left[ k_{TCR>Boost}^{production} . n_{TCR \bullet pMHC_i}(t) + k_{Boost>Boost}^{production} . n_{Boost_i}(t) \right] . \left( n_{Boost_{total}} - n_{Boost_i}(t) \right)$$

$$\frac{dn_{IL2}(t)}{dt} = \frac{N_{cells}}{n_{cells}} \sum_{i=1}^{n_{cell}} \left[ -k_{IL2>IL2R}^{association} . \frac{1}{\mathcal{V} \mathcal{N}_a} . n_{IL2}(t) . \left[ n_{IL2R_i}(t) - n_{IL2 \bullet IL2R_i}(t) \right] \right.$$
$$\left. + \left( k_{TCR>IL2}^{production} + k_{Boost>IL2}^{production} . \frac{n_{Boost_i}(t)}{n_{Boost_{total}}} \right) \times \mathcal{H}\left( \frac{n_{TCR \bullet pMHC_i}(t)}{1 + \sigma_{pSTAT5 \searrow IL2} . \mathcal{P}_{pSTAT5_i}(t)}, \tau_{IL2}^{production} \right) \right]$$

The initial conditions are:

$$n_{pMHC}(t=0) = n_{pMHC}^{total} . \frac{n_{cells}}{\mathcal{N}_{cells}}$$

$$n_{TCR}(t=0) = 3.10^4$$

$$n_{IL2}(t=0) = 0$$

$$n_{IL2R\alpha_i}(t=0) = 0$$

$$n_{IL2R\beta_i}(t=0) = 10^3$$

$$n_{IL2 \bullet IL2R_i}(t=0) = 0$$

$$n_{Boost_i}(t=0) = 0$$

## Parameters

| Parameter | Notation | Value | Reference |
|---|---|---|---|
| Equilibrium constant for pMHC-TCR complex formation | $K_{TCR \bullet pMHC}$ | 30,000 | **Huppa et al., 2003** |
| Equilibrium constant for IL-2R pre-assembly | $K_{IL2R\alpha \bullet IL2R\beta\gamma}$ | 2,700 | Figure S16 in **Cotari et al., 2013** |
| Efficiency of TCR inhibition on STAT5 phosphorylation | $\sigma_{TCR \searrow IL2R}$ | 0.01 | Adjusted to fit **Figure 4A** |
| EC$_{50}$ of conversion of full IL-2• IL-2R into pSTAT5 | $K_{IL2 > pSTAT5}$ | $10^4$ | Adjusted to fit **Figure 4A** |
| Rate of antigen consumption | $k_{antigen}^{consumption}$ | $\ln(2)/3.5\,h^{-1}$ | **von Essen et al., 2004** |
| Threshold number of pMHC–engaged TCR to start activation | $\tau_{pMHC-TCR}^{activation}$ | 1 | **Huang et al., 2013** |
| TCR–dependent IL2Rα expression plateau | $n_{TCR>IL2R\alpha}^{max}$ | 1000 | **Cotari et al., 2013** |
| Rate of internalization for IL2Rα | $k_{receptor}^{degradation}$ | $\ln(2)/5\,h^{-1}$ | **Duprez et al., 1988** |
| pSTAT5–dependent plateau for IL2Rα expression | $n_{pSTAT5>IL2R\alpha}^{max}$ | $2.10^6$ | Figure 1D in **Cotari et al., 2013** |
| Abundance of IL–2Rβ without TCR activation | $n_{IL2R\beta}^{(0)}$ | 1000 | Figure S8 in **Cotari et al., 2013** |
| Maximal abundance of IL2Rβ upon TCR activation | $n_{IL2R\beta}^{max}$ | 10,000 | **Figure 8—figure supplement 1C** |
| Rate of IL-2 binding to full IL-2R | $k_{IL2 \bullet IL2R}^{association}$ | $1\times10^{11}\,h^{-1}$ | Table I in **Wang and Smith, 1987** |
| Rate of internalization of full complex (IL-2• IL-2R) | $k_{IL-2 \bullet IL-2R}^{endocytosis}$ | $\ln(2)/0.25\,h^{-1}$ | **Duprez et al., 1988** |
| TCR-dependent secretion rate of IL-2 | $k_{TCR>IL2}^{production}$ | $7.5\times3600\,h^{-1}$ | **Figure 8—figure supplement 1B** |
| pSTAT5–dependent inhibition of IL2 production | $\sigma_{pSTAT5 \searrow IL2}$ | $3.10^5$ | Fit in this study |
| Total number of Boost molecules | $n_{Boost_{total}}$ | $10^5$ | Fit in this study |
| TCR–dependent activation of IL-2 Boost | $k_{TCR>Boost}^{activation}$ | $10^{-3}$ | Fit in this study |
| Positive feedback on the activation of Boost | $k_{Boost>Boost}^{activation}$ | $3.10^{-1}$ | Fit in this study |
| Boost-dependent secretion rate of IL-2 | $k_{Boost>IL2}^{production}$ | $30 \times k_{TCR>IL2}^{production}$ | **Figure 8—figure supplement 1B** |

## Algebraic relationship between pSTAT5, IL-2Rα and antigen for a mixture of two T cell clones

As experimentally shown in **Figure 10C**, the model predicts that the ratio $\frac{pSTAT5_2 *(IL-2R\alpha)_1}{pSTAT5_1 *(IL-2R\alpha)_2}$ is a function of the ratio $\frac{[Antigen_2]}{[Antigen_1]}$. We anticipated that the TCR-driven inhibition of IL-2 signaling that we characterized experimentally (**Figure 4**) and theoretically (**Figure 9B**) would yield interesting dynamics for the IL-2 pathway in a mixture of T cell clones cultured together. Our quantitative understanding from **Figure 4A** lead to the following equation for each T cell clone $i$ in the culture

$$\# pSTAT5_i = [IL-2] * (\#IL-2R\alpha)_i * f\left([Antigen_i]\right),$$

where $f$ is a function and #X represents the number of X. As the concentration of IL-2 is a shared variable for the two co-cultured T cell clones, we can eliminate it by simple algebra:

$$\frac{\# pSTAT5_2 * (\#IL-2R\alpha)_1}{\# pSTAT5_1 * (\#IL-2R\alpha)_2} = \frac{f\left([Antigen_2]\right)}{f\left([Antigen_1]\right)}.$$

This lead us to plot in *Figure 10C*:

$$\left(\#\,pSTAT5_2 * (\#\,IL-2R\alpha)_1\right) = \mathcal{F}\left([Antigen_1],[Antigen_2]\right) * \left(\#\,pSTAT5_1 * (\#\,IL-2R\alpha)_2\right),$$

where the correlation coefficient $F$ is defined as:

$$\mathcal{F}\left([Antigen_1],[Antigen_2]\right) = \frac{f\left([Antigen_2]\right)}{f\left([Antigen_1]\right)}.$$

We fit the antigen dependency for the median $\dfrac{\#\,pSTAT5_{5C.C7}*(\#\,IL-2R\alpha)_{A1(M)}}{\#\,pSTAT5_{A1(M)}*(\#\,IL-2R\alpha)_{5C.C7}}$ of different time points (*Figure 10—figure supplement 1C*). This formula fits the experimental data in *Figure 10C*, with the additional quantification that this prefactor $\mathcal{F}$ scales with the quantities of $Antigen_1$ and $Antigen_2$:

$$\mathcal{F}\left([Antigen_1],[Antigen_2]\right) \propto [Antigen_1]^{+0.48\pm0.18} * [Antigen_2]^{-0.49\pm0.18},$$

where the error bars correspond to the 95% confidence interval in our nonlinear regression parameters. Thus, the study of the mixture of two clones uncovered an additional scaling for the inhibitory cross-talk between TCR and IL-2 signaling in 5C.C7 and A1(M) T cells:

$$\frac{\#\,pSTAT5_2 * (\#\,IL-2R\alpha)_1}{\#\,pSTAT5_1 * (\#\,IL-2R\alpha)_2} \propto \left(\frac{[Antigen_2]}{[Antigen_1]}\right)^{-0.49\pm0.18}$$

The quality of the fit is $\chi^2 = 0.8$ for N = 9 conditions and 3 parameters.

## Ex vivo quantitation of tumor antigenicity via measurement of TCR inhibition of IL-2 signaling

C57BL/6 mice were injected in the right flank with $10^5$ B16 melanoma cells. Two weeks post-injection, samples of tumors and tumor-draining lymph nodes were harvested, prepared as single-cell suspension and irradiated at 3000 rad, then co-cultured with 40,000 primary CD4+ TRP-1 transgenic T cells. IL-2 accumulation, IL-2Rα expression and STAT5 phosphorylation among activated TRP-1 cells were measured at intervals of approximately 8 hr for 52 hr. In parallel, the same experiments were performed with lymph node and spleen cells of unchallenged C57BL/6 mice pulsed with titrated amounts of TRP-1 peptide (Calibration series). We fit the IL-2 response as:

$$pSTAT5(t) = Slope_{Xtalk}\cdot\left([IL-2](t).\#\,IL-2R\alpha(t)\right) + Background,$$

in order to estimate the extent of antigen-driven inhibition of IL-2 signaling, which manifests as a decrease in $Slope_{Xtalk}$. The antigen-dependency of $Slope_{Xtalk}$ is derived from the data collected in the Calibration Series. This calibration is then applied to back-calculate the antigenicity of the sampled tumors and lymphocytes.

## Acknowledgements

We would like to thank Ron Germain, Rachel Gottschalk, Eric Siggia, Massimo Vergassola and members of the ImmunoDynamics group for useful discussions and comments on the manuscript. This work was supported by NIH R01-AI083408, NIH U54-CA143798, NIH T32-AIO7621 (KT).

## Additional information

### Funding

| Funder | Grant reference number | Author |
| --- | --- | --- |
| National Institutes of Health | R01-AI083408 | Karen E Tkach, Debashis Barik, Guillaume Voisinne, Matthew M Hathorn, Jesse W Cotari, Robert Vogel, Oleg Krichevsky, Grégoire Altan-Bonnet |

| Funder | Grant reference number | Author |
| --- | --- | --- |
| National Institutes of Health | U54-CA143798 | Guillaume Voisinne, Jesse W Cotari, Robert Vogel, Grégoire Altan-Bonnet |
| National Institutes of Health | T32-AIO7621 | Karen E Tkach |

The funders had no role in study design, data collection and interpretation, or the decision to submit the work for publication.

## Author contributions

KET, GA-B, Conception and design, Acquisition of data, Analysis and interpretation of data, Drafting or revising the article; DB, Conception and design, Acquisition of data, Analysis and interpretation of data; GV, Acquisition of data, Analysis and interpretation of data; NM, MMH, JWC, RV, interpreting the data, Acquisition of data, Drafting or revising the article; TM, JW, interpreting the data, Conception and design, Drafting or revising the article; OK, Conception and design, Analysis and interpretation of data

## Ethics

Animal experimentation: This study was performed in strict accordance with the recommendations in the Guide for the Care and Use of Laboratory Animals of the National Institutes of Health. All of the animals were handled according to approved institutional animal care and use committee (IACUC) protocols (#05-031) of Memorial Sloan Kettering Cancer Center (New York NY)

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
