## [Decision Letter]

Thank you for sending your work entitled “Dissecting how T cells translate individual/quantal antigen responses into collective/analog scaling in IL-2 secretion” for consideration at *eLife*. Your article has been favorably evaluated by a Senior editor and 3 reviewers, one of whom is a member of our Board of Reviewing Editors.

The Reviewing editor and the other reviewers discussed their comments before we reached this decision, and the Reviewing editor has assembled the following comments to help you prepare a revised submission.

This is a very interesting paper, where experiments and synergistic computational models are used to understand how IL-2 may regulate population-level responses of T cells. The connections between single cell behavior and that of the population are interesting. The main empirical result is that IL-2 production is weakly dependent on the number of T cells and strongly dependent on antigen amount, and this relationship obeys a scaling law. Since this result does not emerge from the existing models of IL-2 signaling, the authors carry out a careful set of studies to provide new mechanistic insights. They find that an interesting coupling between a coherent feed-forward loop and different time dependence of IL-2 production on antigen quantity and number of T cells can recapitulate the experimental findings. These and other findings reported in this long paper are interesting. We believe this paper is publishable in *eLife*, provided the authors properly address the following issues.

1) In the Results section entitled “Population-size independent scaling of IL-2 accumulation with antigen dose”, it is stated that IL-2 production varied by three orders of magnitude with antigen dose. However, when I look at Figure 1, the ratio of the magnitudes at the peaks (filled points) appears to be no more than 300-fold. This makes no qualitative difference to the authors' arguments, but what did I read wrong in Figure 1?

2) The scaling law reported in [Disp-formula equ1] is a very nice result. On a critical note, however, the primary experimental evidence of the scaling laws is presented in a less than optimal manner (Figure 1). It is pretty hard to really know what is going on with the three dimensional plot in Figure 1, as the depth of deviation from power law is difficult to see. It seems important to present the scaling laws so critical to the rest of the paper via improved figures and data analyses. Perhaps 2-dimensional projections that clearly show the slope of the fitted line (exponent) on logarithmic coordinates, as is typically done in the physics literature, might be appropriate.

3) [Disp-formula equ2] does not seem to predict saturation of antigen dependence. Yet, it is stated that the known model predicts saturation at 10 pm. If indeed the model does predict saturation (not clear from [Disp-formula equ2]), it would be good to give a qualitative reason as to why the topology of the network results in saturation.

4) In the Results section entitled “Computational model of the IL-2 pathway demonstrates the significance of experimentally characterised feedbacks”, the fact that the computational model recovers qualitative features based on the topology of the network is a nice result. But, the authors go on to note that it predicts the scaling exponents (and other features) in a fairly quantitative manner. This raises the following important question. Are all the parameters in the complex model shown in Figure 4 known? For example, how can the parameters associated with the phenomenological “Boost” variable's time-dependent activation be known from experiments? This important point needs clarification.

5) Also in the Results section entitled “Computational model of the IL-2 pathway demonstrates the significance of experimentally characterised feedbacks”, while there is nothing wrong with using differential equations for this model, some commentary on why stochastic effects are unimportant (e.g., large T cell populations) should be noted.

6) The equation following the line “Simple algebra (Figure S5E) demonstrates how the cue-signal-response to IL-2 in a mixed population of cells predicts the distribution of stimulating antigens” suggests that the ratio on the left should be 1 if antigen 1 and antigen 2 are presented in equal amounts, since the function, f, on the right is the same for both antigens. It is difficult to tell from Figure 6 whether, for the cases corresponding to the diagonal elements of the Tables below, the slope of the curve is indeed 1. If not, how different are they from unity? How do the experiments and model compare for these cases? This is important to note in order to support the claim that the model predictions follow the equation, and that the experiments mirror this.

7) Finally, a non-technical point that the authors might wish to consider is including a few more references for completeness. Regarding digital signaling in T cells, perhaps, the contribution from the Weiss and Chakraborty labs (e.g., Cell, 2009) may also be noted? Several scientists have used computational models fruitfully to study other aspects of immunology, such as host-pathogen dynamics (e.g., Perelson, deBoer labs), constraints on viral mutability and vaccine design (e.g., Chakraborty lab), and characterization of antibody and T cell repertoires (e.g., Quake, Callen labs). In the Introduction where the authors describe the value of computational studies in immunology, it may be worth mentioning such studies so as to give a complete picture of the role of computation and theory. Mentioning studies that span scales is especially significant, as in their paper, the authors span scales in a nice way.

---

## [Author Response]

*1) In the Results section entitled “Population-size independent scaling of IL-2 accumulation with antigen dose”, it is stated that IL-2 production varied by three orders of magnitude with antigen dose. However, when I look at*
Figure 1*, the ratio of the magnitudes at the peaks (filled points) appears to be no more than 300-fold. This makes no qualitative difference to the authors' arguments, but what did I read wrong in*
Figure 1?

We originally stated: “Strikingly, [IL-2]_max_ varied by more than three orders of magnitude with antigen dose, despite the aforementioned limited dynamic range of early T cell responses.” This sentence was confusingly worded; “over three orders of magnitude” was meant to describe the input dynamic range of resolution (i.e., our capacity to tell different antigen doses apart), and not the dynamic range of IL-2 output. We have changed this sentence for better clarity: “Strikingly, [IL-2]_max_ scaled as a power law that could distinguish more than three orders of magnitude of input antigen dose, despite the aforementioned saturated dynamic range of early T cell responses.”

*2) The scaling law reported in*
[Disp-formula equ1]
*is a very nice result. On a critical note, however, the primary experimental evidence of the scaling laws is presented in a less than optimal manner (*Figure 1*). It is pretty hard to really know what is going on with the three dimensional plot in*
Figure 1*, as the depth of deviation from power law is difficult to see. It seems important to present the scaling laws so critical to the rest of the paper via improved figures and data analyses. Perhaps 2-dimensional projections that clearly show the slope of the fitted line (exponent) on logarithmic coordinates, as is typically done in the physics literature, might be appropriate*.

We concur with the reviewers that these additional graphs are necessary to better represent the newly found scaling laws. In Figure 1, we included the two-dimensional plots demonstrating the scaling for [IL-2]_max_ as a function of antigen quantity for different numbers of cells, or as a function of population size for varied quantities of input antigen. In Figure 1, we have combined experiments in which T cells were stimulated with 1, 2, or 2.5 * 10^-6^/10^-7^/10^-8^/10^-9^ M antigen, and thus indicate that the plots are segregated according to the order of magnitude rather than exact amount of antigen. We appreciate that the expansive format of *eLife* enables such a complete representation of our experimental results through these two additional panels (Figure 1).

*3)*
[Disp-formula equ2]
*does not seem to predict saturation of antigen dependence. Yet, it is stated that the known model predicts saturation at 10 pm. If indeed the model does predict saturation (not clear from*
[Disp-formula equ2]*), it would be good to give a qualitative reason as to why the topology of the network results in saturation*.

To address this question, we have included a figure illustrating the [IL-2] maxima predicted by simulations of the “Classical Model” for a wide range of input conditions (Figure 1). This figure more clearly illustrates the saturated output result than Figure S2C, which highlighted the dynamics of only a few cases. [Disp-formula equ2] represents the results of a PLSR fit across the simulation outputs shown in Figure 1. As the reviewer pointed out, the equation does not capture the saturation, but reports the predicted population size dependence and shallower resolution of input antigen dose that emerged from the “Classical Model”. We also revised our explanation of why prior understanding of IL-2 pathway topology predicted saturation at 10 pM—and how this prediction differs from our experimental results—for thoroughness and clarity (paragraph starting “ Furthermore, this model of IL-2 production predicted a weak antigen dependency that saturated near the canonical IL-2 signaling threshold of 10 pM…).

*4) In the Results section entitled ‘Computational model of the IL-2 pathway demonstrates the significance of experimentally characterised feedbacks”, the fact that the computational model recovers qualitative features based on the topology of the network is a nice result. But, the authors go on to note that it predicts the scaling exponents (and other features) in a fairly quantitative manner. This raises the following important question. Are all the parameters in the complex model shown in*
Figure 4
*known? For example, how can the parameters associated with the phenomenological “Boost” variable's time-dependent activation be known from experiments? This important point needs clarification*.

A key aspect of our study is the systematic effort to experimentally parameterize our computational model. In Figure 4, we represented all biochemical parameters that were derived from experimental data in a green font. We were left with 4 parameters (represented in a black font) that were phenomenologically derived from our experiments. These are associated with the two new feedbacks that were uncovered in this study: the TCR-IL-2R crosstalk and the boost in IL-2 secretion. The molecular mechanisms underlying these experimentally characterized network connections are currently unknown. However, the architecture we chose and parameters we fit to describe these mechanisms were heavily constrained by both the large set of known system parameters, and the quantitative experimental dynamics that served as fitting targets.

In the case of the “Boost”, we have added a more thorough explanation of our rationale in choosing a TCR-driven feedback, derived from our and others’ experiments (paragraph starting “In modelling the time-dependent acceleration in IL-2 secretion, we followed several lines of evidence…”

*5) Also in the Results section entitled ‘Computational model of the IL-2 pathway demonstrates the significance of experimentally characterised feedbacks”, while there is nothing wrong with using differential equations for this model, some commentary on why stochastic effects are unimportant (e.g., large T cell populations) should be noted*.

Indeed, most chemical reactions in our model involve large number of molecules, such that the implementation of stochastic chemical equation was deemed unnecessary. For example, the number of STAT5 transcription factors is in the order of 3000 molecules, such that number fluctuations would be in the order of sqrt(3000) = 54, hence 2 % of the molecules. Thus, we estimate that stochastic fluctuations would add limited corrections to our model simulated with ordinary differential equations.

We included a justification of this deterministic simplification of our model based on insights from other studies. Stochasticity in chemical reactions has been shown to be essential in biological networks where timing imbalance in feedback regulation generates a switch, or in cases where the onset of responses depends on proteins in very low abundance (such as in regulation of cell cycle or TCR triggering). As these considerations do not apply in our model, we elected to use deterministic ordinary differential equations at present time.

If there exist stochastic effects, in particular in terms of regulation of gene expression, one would need to coarse-grain our model to allow rapid testing of the relevance of stochasticity in the functional outcome of our simulations. This is particularly relevant when modeling the distributions of expression of receptors, cytokine and signaling responses – an issue beyond the scope of this study. Ongoing work is addressing these issues and will be the subject of a subsequent publication.

*6) The equation following the line “Simple algebra (**Figure S5E**) demonstrates how the cue-signal-response to IL-2 in a mixed population of cells predicts the distribution of stimulating antigens” suggests that the ratio on the left should be 1 if antigen 1 and antigen 2 are presented in equal amounts, since the function, f, on the right is the same for both antigens. It is difficult to tell from*
Figure 6
*whether, for the cases corresponding to the diagonal elements of the Tables below, the slope of the curve is indeed 1. If not, how different are they from unity? How do the experiments and model compare for these cases? This is important to note in order to support the claim that the model predictions follow the equation, and that the experiments mirror this*.

Thank you for this suggestion. We revisited the experimental and computational results in Figure 6. As pointed out by the reviewer, our model predictions demonstrated a small deviation in the ratios of (#pSTAT5)*(# IL-2R α) for equal ratio of antigen (Figure 6 - left). This result was unexpected, as both clones were assigned identical equations and parameters. However, our model introduces some stochasticity in the activation times of clones 1 and 2, which are chosen from a uniform distribution to emulate the un-synchronized nature of T cell-APC encounters. Further inquiry of our computational results demonstrated that such randomness in the activation times was sufficient to generate a slight discrepancy in the IL-2 responses when simulating small numbers of cells. Re-running our model with a larger number of cells (50 instead of 20) alleviated the discrepancy in #pSTAT5*#IL-2R α products as presented in our new Figure 6 – left.

Following the reviewers’ suggestion, we added a dashed isoline in our model predictions for better examination of symmetry between clones activated with equal antigen doses. One factor that could contribute to asymmetrical alignment is differential affinity between each clone’s TCR and cognate antigen. Strikingly, these trajectories are indeed symmetrical in our experimental results, which is particularly surprising since we do not control the potency of the K5 and HY antigens for 5C.C7 and A(1)M clones. On the other hand, both antigens have been documented to be particularly strong; thus, it is logical that our experimental results reproduce the symmetrical trajectories predicted by our computational model.

*7) Finally, a non-technical point that the authors might wish to consider is including a few more references for completeness. Regarding digital signaling in T cells, perhaps, the contribution from the Weiss and Chakraborty labs (e.g., Cell, 2009) may also be noted? Several scientists have used computational models fruitfully to study other aspects of immunology, such as host-pathogen dynamics (e.g., Perelson, deBoer labs), constraints on viral mutability and vaccine design (e.g., Chakraborty lab), and characterization of antibody and T cell repertoires (e.g., Quake, Callen labs). In the Introduction where the authors describe the value of computational studies in immunology, it may be worth mentioning such studies so as to give a complete picture of the role of computation and theory. Mentioning studies that span scales is especially significant, as in their paper, the authors span scales in a nice way*.

We concur, and have expanded our introduction to highlight more recent successes in quantitative immunology. We are grateful that the expansive format of *eLife* to allow us to include a more comprehensive introduction of other modeling efforts in immunology.